



# A convolution of observational and model data to estimate age of air spectra in the northern hemispheric lower stratosphere

Marius Hauck[1], Harald Bönisch[5], Peter Hoor[4], Timo Keber[1], Felix Ploeger[2,3], Tanja J. Schuck[1], and Andreas Engel[1]

[1]Institute for Atmospheric and Environmental Sciences, Goethe University Frankfurt am Main, Frankfurt am Main, Germany
[2]Institute for Energy and Climate Research: Stratosphere (IEK-7), Forschungszentrum Jülich, Jülich, Germany
[3]Institute for Atmospheric and Environmental Research, University of Wuppertal, Wuppertal, Germany
[4]Institute for Atmospheric Physics, Johannes Gutenberg University Mainz, Mainz, Germany
[5]Karlsruhe Institute of Technology, Institute of Meteorology and Climate Research – Atmospheric Trace Gases and Remote Sensing, Eggenstein-Leopoldshafen, Germany

*Correspondence to*: Marius Hauck (hauck@iau.uni-frankfurt.de)

**Abstract.** Derivation of mean age of air (AoA) and age spectra from atmospheric measurements remains a challenge and often requires data from atmospheric models. This study tries to minimize the direct influence of model data and presents an extension and application of a previously established inversion method to derive age spectra from mixing ratios of long- and short-lived trace gases. For a precise description of cross-tropopause transport processes, the inverse method is extended to incorporate air entrainment into the stratosphere across the tropical and extratropical tropopause. We first use simulations with the Chemical Lagrangian Model of the Stratosphere (CLaMS) to provide a general proof of concept of the extended principle in a controllable and consistent environment, where the method is applied to an idealized set of ten trace gases with predefined constant lifetimes and compared to reference model age spectra. In the second part of the study we apply the extended inverse method to atmospheric measurements of multiple long- and short-lived trace gases measured aboard the High Altitude and Long Range (HALO) research aircraft during the two research campaigns POLSTRACC/GW-LCYCLE/SALSA (PGS) and Wave-driven Isentropic Exchange (WISE). As some of the observed species undergo significant loss processes in the stratosphere, a Monte Carlo simulation is introduced to retrieve age spectra and chemical lifetimes in stepwise fashion and to account for the large uncertainties. Results show that in the idealized model scenario the inverse method retrieves age spectra robustly on annual and seasonal scale. The extension to multiple entry regions proves reasonable as our CLaMS simulations reveal that in the model between 50 % and 70 % of air in the lowermost stratosphere has entered through the extratropical tropopause (30° – 90° N/S) on annual average. When applied to observational data of PGS and WISE the method derives age spectra and mean AoA with meaningful spatial distributions and quantitative range, yet large uncertainties. Results indicate that entrainment of fresh tropospheric air across both extratropical and tropical tropopause has peaked prior to both campaigns, but with lower mean AoA for WISE than PGS data. For a full assessment the ratio of moments for all retrieved age spectra is evaluated and found to range between 0.52 years and 2.81 years for PGS and WISE. It is concluded that the method derives reasonable and consistent age spectra using observations of chemically active trace gases. Our findings might contribute to an improved assessment of transport with age spectra in future studies.





## 1 Introduction

Spatial distributions of many greenhouse gases and ozone-depleting trace gases throughout the stratosphere are steered by a global mean meridional circulation, known as Brewer-Dobson circulation (BDC), making it a crucial factor for the Earth's radiative budget and climate (Shepherd, 2007; Solomon et al., 2010). The BDC is usually characterized as a superposition of a mean residual circulation with net mass transport and two-way eddy mixing with tracer exchange but no net mass flux (Plumb, 2002; Butchart, 2014). Birner and Bönisch (2011) recognize two distinct pathways for the BDC, a shallow and a deep branch, with different transport time scales along them. The shallow branch reaches from the tropics into the extratropics close to the tropopause, while the deep branch extends up into the middle and upper stratosphere (Birner and Bönisch, 2011). Mechanical drivers of the BDC are planetary- and synoptic-scale atmospheric waves that get excited in the troposphere, propagate upward into the extratropical middle stratosphere where they finally break and transfer their momentum to induce a poleward motion (Haynes et al., 1991; Holton et al., 1995). The wave drag causes air to rise slowly in the tropics mainly through the tropical tropopause layer (TTL) to compensate the poleward drift (Fueglistaler et al., 2009). Eventually, air descends at higher latitudes back into the troposphere. The upward mass flux in the tropics succumbs a distinct seasonality with maximum upward transport during northern hemispheric (NH) winter, due to a maximum of tropospheric waves (Rosenlof and Holton, 1993; Rosenlof, 1995). Although the TTL is identified as the main entry point to the stratosphere, isentropic transport across the extratropical tropopause plays an important role for air composition in the lowermost stratosphere (LMS) below 380 K potential temperature (Olsen et al., 2004; Boothe and Homeyer, 2017). Those exchange processes exhibit their own distinct seasonality in each hemisphere (Appenzeller et al., 1996; Škerlak et al., 2014).

As global greenhouse gas concentrations and sea surface temperatures keep rising, model studies expect that the BDC generally strengthens in consequence of an enhanced wave drag (Garcia and Randel, 2008; Li et al., 2008; Shepherd and McLandress, 2011). Studies of suitable dynamical tracers (e.g., $SF_6$, $CO_2$ or $N_2O$) from different observational sources, however, show a much more complex and contradictory state indicating that the strength of the BDC might undergo nonuniform structural changes with hemispheric asymmetries (e.g., Engel et al., 2009; Bönisch et al., 2011; Ray et al., 2014; Stiller et al., 2017; Laube et al., 2020). Although more recent analyses of global models (e.g., Oberländer-Hayn et al., 2015; Oberländer-Hayn et al., 2016) and also reanalyses (e.g., Diallo et al., 2012; Abalos et al., 2015) were able to disentangle some inconsistencies, possible trends of the BDC remain an open issue. Especially in case of reanalyses, as recent studies show that different reanalysis products can alter the outcome significantly (Chabrillat et al., 2018; Ploeger et al., 2019).

A major problem that studies of the BDC share is the difficulty to measure transport directly (Butchart, 2014). While model simulations provide possibilities to derive quantities that describe strength and structure of the BDC and potential trends, observational analysis is challenging, especially in remote parts of the stratosphere where only sparse measurements exist. A well-established diagnostic tool used in many studies of both models and observations is mean age of air (AoA) (Hall and Plumb, 1994). Mean AoA is defined as the average transit time an air parcel needs to reach the considered location starting at a specified reference surface, usually the Earth's surface or the tropical tropopause. It is linked inversely proportional to the





general circulation strength (Austin and Li, 2006). Mean AoA is also influenced by mixing processes (Waugh and Hall, 2002; Garny et al., 2014) and separation between residual transport and mixing is complicated due to the average nature of mean AoA. For such analysis, a full transit time distribution should be considered, since stratospheric air consists of an irreversible mixture of air parcels with different transit times from the source region. The age spectrum of any arbitrary air parcel represents

a probability density function (PDF) of the transit time scales within the parcel (Kida, 1983).

In many model simulations, the age spectrum is constructed by an implementation of chemically inert trace gases that are periodically pulsed in a specified boundary region (e.g., Haine et al., 2008; Li et al., 2012; Ploeger and Birner, 2016). Mean AoA is then defined as the first moment of the age spectrum. In case of observations, the derivation of both mean AoA and age spectra is more complex and follows different approaches. Basis of many studies in the past have been measurements of

(very) long-lived trace gases together with the fundamental theory on age spectra by Hall and Plumb (1994) to constrain the shape of the spectra and the ratio of variance to mean AoA beforehand (e.g., Volk et al., 1997; Engel et al., 2002; Engel et al., 2009). Recent results by Fritsch et al. (2019) show that the parameter choice in such constraint methods strongly influences resulting mean AoA trends. Other methods rely on a more general shape of the spectrum, but require more data than in the constrained case (Holzer and Primeau, 2010; Holzer and Waugh, 2015). However, the number of suitable stratospheric trace

gases with (very) large chemical lifetime is limited. One possible solution is to consider additionally substances with rapid chemical depletion, since stratospheric chemistry and transport are strongly intertwined. Such approaches also exist in different constrained (e.g., Schoeberl et al., 2005; Ehhalt et al., 2007) and unconstrained versions (e.g., Schoeberl et al., 2000; Podglajen and Ploeger, 2019). An improved parametric approach has been introduced in Hauck et al. (2019), which relies only on a constrained age spectrum shape to achieve applicability together with well-matching results in a model test scenario.

Unfortunately, the method shows quite large discrepancies in the lowermost stratosphere where most stratospheric aircraft measurements are taken.

This paper constitutes a direct follow-up to Hauck et al. (2019). We extend the therein described inverse method to the lowermost stratosphere with a new formulation and provide a short proof of concept using a simulation of the Chemical Lagrangian Model of the Stratosphere (CLaMS) (McKenna, 2002a, 2002b; Pommrich et al., 2014) with idealized radioactive

tracers. We then apply the extended method to in situ measurement data gained during the campaigns POLSTRACC / GW-LCYCLE / SALSA (PGS) and Wave-driven Isentropic Exchange (WISE) of the High Altitude and Long Range (HALO) research aircraft and analyze the resulting age spectra and their moments. Section 2 gives insight into the extended formulation of the method and the statistical procedure to estimate age spectra and chemical lifetimes from observations. Section 3 describes the data basis for this study. Finally, results are presented in Sect. 4 and completed by an outlook and a critical

discussion in Sect. 5.





## 2 Methodology

### 2.1 Inverse method – general approach and problems

The theory of the inverse method is provided in detail by Hauck et al. (2019). It is a modified version of the method presented by Schoeberl et al. (2005) and utilizes mixing ratios of a set of different chemically active compounds to derive an age spectrum

5   using a numerical optimization scheme. The age spectrum shape is constrained by the inverse Gaussian distribution proposed by Hall and Plumb (1994). At the same time, seasonality in stratospheric transport, which is visible in modelled age spectra in the form of multiple modes (e.g., Reithmeier et al., 2008; Li et al., 2012; Ploeger and Birner, 2016), has to be imposed by a seasonal scaling factor as the inverse Gaussian function is intrinsically a monomodal PDF. Mathematically, the inverse method is based on the following equation

$$\chi(\vec{x},t) = \int_0^\infty \chi_0(t-t') \cdot e^{-\frac{t'}{\tau(\vec{x},t,t')}} \cdot G(\vec{x},t,t') \cdot \omega(t') \cdot n(\vec{x},t) \cdot dt'. \qquad (1)$$

$\chi(\vec{x},t)$ denotes the mixing ratio of any arbitrary trace gas at $(\vec{x},t)$ in the stratosphere with chemical depletion, but no stratospheric sources. $t'$ is the transit time through the stratosphere, $\chi_0(t-t')$ the mixing ratio time series of the substance at the reference surface, $\tau(\vec{x},t,t')$ the transit-time-dependent chemical lifetime, $G(\vec{x},t,t')$ the age spectrum and $\omega(t')$ the

15   seasonal scaling factor to gain multimodal PDFs (see below). $n(\vec{x},t)$ is a normalization factor for the age spectrum, which ensures that $G(\vec{x},t,t')$ and $G(\vec{x},t,t') \cdot \omega(t')$ have identical norms. It is defined as

$$n(\vec{x},t) = \frac{\int_0^\infty G(\vec{x},t,t') \cdot dt'}{\int_0^\infty G(\vec{x},t,t') \cdot \omega(t') \cdot dt'}. \qquad (2)$$

The definition of $n(\vec{x},t)$ above preserves the norm of the age spectrum $G(\vec{x},t,t')$ during the scaling process. Although age

20   spectra must usually be normalized, there are cases in this study where an non-normalized spectrum is physically meaningful (see Sect. 2.2.1). Full consideration of a transit-time-dependent entry mixing ratio has been introduced for applicability purposes, since many atmospheric trace gases exhibit a strong long-term temporal trend that should be considered properly. An approximation of the stratospheric age spectrum in Eq. ( 1 ) is provided by Hall and Plumb (1994)

$$G(\vec{x},t,t') = \frac{z}{2\sqrt{\pi K(\vec{x},t)t'^3}} \cdot e^{\left(\frac{z}{2H} - \frac{K(\vec{x},t)t'}{4H^2} - \frac{z^2}{4K(\vec{x},t)t'}\right)}, \qquad (3)$$



but with a three-dimensional transport parameter $K(\vec{x}, t)$ instead of an originally one-dimensional diffusion coefficient. $z$ is the potential temperature difference to the local tropopause and $H$ the scale height of the air density. The first moment $\Gamma(\vec{x}, t)$ (i.e., mean AoA) and centered second moment $\Delta^2(\vec{x}, t)$ (i.e., variance) of the spectrum are given as (Hall and Plumb, 1994)

$$\Gamma(\vec{x}, t) = \int_0^\infty G(\vec{x}, t, t') \cdot t' \cdot dt', \tag{4}$$

$$\Delta^2(\vec{x}, t) = \frac{1}{2} \cdot \int_0^\infty G(\vec{x}, t, t') \cdot (t' - \Gamma(\vec{x}, t))^2 \cdot dt'. \tag{5}$$

For the inversion process, mixing ratios of a given set of distinct trace gases are considered and $K(\vec{x}, t)$ is optimized numerically for all species simultaneously. Hauck et al. (2019) provide a general proof of concept of this method in a controllable model environment featuring a set of several artificial radioactive trace gases with constant chemical lifetimes.

Despite the robust performance of the inverse method compared to the model reference in general, the lower stratosphere proves challenging especially during northern hemispheric spring and fall. That is most probably linked to a conceptual flaw in the design of the inverse method. In its presented form, all derived inverse age spectra assume the tropical tropopause as single source region into the stratosphere. Although this appears valid for the upper and middle stratosphere (Fueglistaler et al., 2009), studies have shown that for the lowermost extratropical stratosphere quasi-isentropic transport across the local

tropopause has critical influence and strongly steers trace gas burdens in that region (Hoor et al., 2005; Bönisch et al., 2009). Therefore, the assumption of single entry through the tropical tropopause layer is insufficient to estimate a precise age spectrum and must be modified to include air entrainment through the complete tropopause together with related seasonality.

## 2.2 Inverse method with multiple entry sections

### 2.2.1 Concept

To fully incorporate transport processes in the extratropical lowermost stratosphere, an extension of the methodology to multiple entry regions is required. A rather intuitive way to divide the tropopause is the partitioning into a northern (index $N$), a tropical (index $T$) and a southern (index $S$) section, each with a separate age spectrum assigned to them. All entry regions must then add up geographically to span the global tropopause. A mathematically strict derivation of age spectra for several different source regions is given by Holzer and Hall (2000). The ansatz is similar to Bönisch et al. (2009), but with a split of

the reference surface rather than a separation into a tropospheric and a stratospheric fraction. Due to mass conservation, this concept translates into a composite age spectrum by

$$G(\vec{x}, t, t') = g_N(\vec{x}, t, t') + g_T(\vec{x}, t, t') + g_S(\vec{x}, t, t'), \tag{6}$$



with $g_i(\vec{x}, t, t')$ being the age spectrum referring transport to the tropopause section $i$. Since the normalization of $G(\vec{x}, t, t')$ must hold also in the extended case, the integration of $G(\vec{x}, t, t')$ now yields

$$\int_0^\infty G(\vec{x}, t, t') \cdot dt' = \int_0^\infty g_N(\vec{x}, t, t') \cdot dt' + \int_0^\infty g_T(\vec{x}, t, t') \cdot dt' + \int_0^\infty g_S(\vec{x}, t, t') \cdot dt'$$
$$:= f_N(\vec{x}, t) + f_T(\vec{x}, t) + f_S(\vec{x}, t) = 1.$$
( 7 )

5    The lowercased $g_i(\vec{x}, t, t')$ indicates a non-normalized age spectrum and $f_i(\vec{x}, t)$ its respective norm. In terms of transport, the norm $f_i(\vec{x}, t)$ provides an estimate of the fraction of air at $(\vec{x}, t)$, which has entered the stratosphere through tropopause region $i$. The norms are referred to as origin fractions and provide an important toolset for an analysis of seasonality in air entrainment (see Sect. 4.1). Any non-normalized age spectrum can be converted into a proper PDF by division through its origin fraction

$$G_i(\vec{x}, t, t') = \frac{g_i(\vec{x}, t, t')}{f_i(\vec{x}, t)}.$$
( 8 )

Since each $g_i(\vec{x}, t, t')$ constitutes a description of transport for the percentage of air at $(\vec{x}, t)$ that entered the stratosphere through tropopause region $i$, it can therefore also be utilized to calculate the mixing ratio fraction $\chi_i(\vec{x}, t)$ associated with that respective entrainment. Multiplication of Eq. ( 7 ) with $\chi(\vec{x}, t)$ includes those mixing ratio fractions into the new concept

$$\chi(\vec{x}, t) = f_N(\vec{x}, t) \cdot \chi(\vec{x}, t) + f_T(\vec{x}, t) \cdot \chi(\vec{x}, t) + f_S(\vec{x}, t) \cdot \chi(\vec{x}, t) := \chi_N(\vec{x}, t) + \chi_T(\vec{x}, t) + \chi_S(\vec{x}, t),$$
( 9 )

where each individual $\chi_i(\vec{x}, t)$ can be derived from $g_i(\vec{x}, t, t')$ and Eq. ( 1 ) as

$$\chi_i(\vec{x}, t) = \int_0^\infty \chi_{0,i}(t - t') \cdot e^{-\frac{t'}{\tau_i(\vec{x}, t, t')}} \cdot g_i(\vec{x}, t, t') \cdot \omega_i(t') \cdot n_i(\vec{x}, t) \cdot dt'.$$
( 10 )

20    The parameterization provided by Eq. ( 3 ) returns an invariably normalized inverse Gaussian function so that Eq. ( 10 ) must be modified with Eq. ( 8 ) to yield correctly

$$\chi_i(\vec{x}, t) = f_i(\vec{x}, t) \cdot \int_0^\infty \chi_{0,i}(t - t') \cdot e^{-\frac{t'}{\tau_i(\vec{x}, t, t')}} \cdot G_i(\vec{x}, t, t') \cdot \omega_i(t') \cdot n_i(\vec{x}, t) \cdot dt'.$$
( 11 )





Each entry region is now treated with a single entry mixing ratio time series $\chi_{0,i}(t - t')$, a transit-time-dependent lifetime $\tau_i(t')$, a normalized age spectrum $G_i(\vec{x}, t, t')$ and an imposed seasonal cycle $\omega_i(t')$ with its respective normalization factor $n_i(\vec{x}, t)$. The introduced formulation is valid for any partitioning of the tropopause into three sub-regions. For this study, the tropical tropopause is chosen to range from 30° S up to 30° N to incorporate the seasonal shift of the intertropical convergence

zone (ITCZ). The northern and southern parts extent from 30° N to 90° N and 30° S to 90° S respectively. With that choice all entry regions span an identical range of 60° latitude. Transport is now characterized by three separate parameters $K_i(\vec{x}, t)$. Since the mixing ratio fraction $\chi_i(\vec{x}, t)$ is usually unknown for any stratospheric location, Eq. ( 11 ) is divided by $f_i(\vec{x}, t)$ knowing that $\chi_i(\vec{x}, t)$ has been introduced as $\chi(\vec{x}, t) \cdot f_i(\vec{x}, t)$. This important step yields a set of three decoupled equations, which can be treated separately

$$\chi(\vec{x}, t) = \int_0^\infty \chi_{0,i}(t - t') \cdot e^{-\frac{t'}{\tau_i(\vec{x}, t, t')}} \cdot G_i(\vec{x}, t, t') \cdot \omega_i(t') \cdot n_i(\vec{x}, t) \cdot dt'. \tag{12}$$

The inversion process of Eq. ( 12 ) is independent of $f_i(\vec{x}, t)$, but only works correctly if $\chi_{0,i}(t - t')$ or $\tau_i(t')$ are unequal for all tropopause regions. Otherwise, each inversion leads to the identical age spectrum. Due to refinements of the optimization algorithm, those equations are optimized for all considered species at once now with 0.1 % tolerance and the new metric of

the symmetric signed percentage bias (SSPB) (Morley et al., 2018). The SSPB is suitable if the overall quantitative range of mixing ratios is large and utilizes the median and logarithm to smooth the strong percentage influence of very small mixing ratios. That inversion process is now called $K$-inversion. Note that the decision to optimize bias rather than variance, e.g., root mean square error, has been made to return an average age spectrum that captures even fine effects of the underlying mixing ratio data robustly. In return, this comes at the cost of higher variance around the true solution due to the bias-variance tradeoff.

**2.2.2 Extratropical seasonal cycles**

Exchange processes across the northern and southern extratropical tropopause each display a different seasonality than the transport through the tropical tropopause layer. These seasonal cycles are relevant for stratospheric age spectra as they cause the multimodal shape of modelled spectra. The inverse method rests upon the monomodal inverse Gaussian function by Hall and Plumb (1994) so that multiple peaks are not included intrinsically but are imposed by the fixed scaling factor $\omega(t')$ during

the inversion process. In Hauck et al. (2019) it has been shown that the ratio of the tropical net upward mass flux at 70 hPa between different seasons can be used to scale the age spectrum at matching transit times. The scaling factor by Hauck et al. (2019) is given as

$$\omega_i(t') = A_i + B_i \cdot \cos\left(\frac{2\pi}{365 \text{ days}} \cdot t' + C_i\right). \tag{13}$$





$A_i$, $B_i$ and $C_i$ are constants that now depend on the entry region and the considered season. Seasons are hereafter abbreviated as DJF (northern winter), MAM (northern spring), JJA (northern summer), and SON (northern fall). The values for the tropical tropopause section are taken from Hauck et al. (2019) and given in the first three columns of Tab. 1. The extratropical cycles are more challenging. According to Fig. 6 in Appenzeller et al. (1996), the flux across the northern and southern tropopause

changes its direction throughout the year and the varying sign thus inhibits a straightforward scaling as for the tropics. For example, if the ratio of spring (downward flux with negative sign) and fall (upward flux with positive sign) was considered for the northern hemisphere, the division of both fluxes in these seasons would lead to a negative scaling factor and hence to indefinite negative values in the age spectrum. To retrieve a correct scaling factor, a net directional flux must be considered. Olsen et al. (2004) provide the net downward mass flux across the northern and southern extratropical 380 K isentropic surface

in their Fig. 1 (hereafter named Olsen flux). This downward motion should be coupled inversely to the flux across the tropopause, exerting a similar forcing as the downward control principle (Haynes et al., 1991). The minimal downward Olsen flux is visible in June. Note that minimum refers to the point closest to zero, since we specify the flux as being downward. This correlates well with the transport across the northern tropopause in Fig. 6 of Appenzeller et al. (1996), that starts to turn upward rapidly in June and reaches its upward maximum with some time lag in late September. Consistently, the maximum

downward Olsen flux is visible in late January, where the flux across the northern tropopause turns also downward and attains its downward maximum in May with a similar delay of three months. The time lag appears equivalent for minimum and maximum, as it takes some time for the signal to propagate from the 380 K surface down to the tropopause. The inverse link between 380 K and tropopause flux is also corroborated by Hoor et al. (2005) and Bönisch et al. (2009), who find a flushing of the northern lowermost stratosphere with fresh tropospheric air during northern summer when the downward Olsen flux is

minimal. An analog feedback is detected in the southern hemisphere, but with six months offset. The less pronounced cycle in the southern hemisphere of Appenzeller et al. (1996) is not visible in the Olsen flux as the cycles of both hemispheres appear similarly strong. This coincides with Škerlak et al. (2014), who also find an alike seasonality of the troposphere to stratosphere transport in both hemispheres. A proper scaling factor for the northern and southern tropopause section is created by turning the seasonal mean Olsen flux into reciprocal values and performing the established relative scaling of all seasons, for instance

spring relative to fall. In that way, the maximum is located for both hemispheres in late spring and the minimum vice versa in late fall.

When the flux ratios are inserted into Eq. ( 13 ), the obtained equations can be solved to retrieve $A_i$, $B_i$ and $C_i$. The coefficients for the northern and southern tropopause are shown in Tab. 1 and the factors are illustrated for all seasons and hemispheres in Fig. 1 for one year transit time. The northern (panel (a)) and southern (panel (b)) factors seem to mirror the seasonality of the

upward flux correctly. The maximum is located at transit times that resemble late spring in all curves (e.g., at 0.6 years for DJF in panel (a), which corresponds to late May or early June). The minimum is then found accordingly at transit times for late fall (e.g., at 0.65 years for DJF in panel (b), which is also equivalent to late May). Just as for the tropics, all cycles are designed so that no scaling occurs in the season they depict, i.e., at transit times 0 and all integer multiples of 1 year. The





amplitudes of the scaling factors come out quite identical in both hemispheres except for fall (orange in **(a)**, green in **(b)**), which undergoes stronger scaling in the south.

### 2.2.3 Limitations

The extended inverse ansatz keeps the benefit of an inversion with a single parameter, but it also holds some disadvantages that require the use of data from atmospheric transport models. The goal of this study is to reduce the amount of necessary modelled data as much as possible for the inversion and evaluation of age spectra, but it must be stated clearly that it is not feasible to provide a method based solely on observations. While all $G_i(\vec{x}, t, t')$ can be well retrieved and evaluated to investigate the BDC without explicit use of the origin fractions $f_i(\vec{x}, t)$ from models, they must be known beforehand to calculate the composite age spectrum $G(\vec{x}, t, t')$ as superposition of all $G_i(\vec{x}, t, t')$. This piece of information must be provided by a model simulation. Also, the choice of a prescribed inverse Gaussian function might not be valid for any point in the stratosphere, particularly for cross-hemispheric PDFs (i.e., $G_S(\vec{x}, t, t')$ in the northern hemisphere and vice versa). However, it is expected that the fractions of interhemispheric exchange are vanishingly low in the lowermost stratosphere making those age spectra negligible. For the remaining distributions it is assumed that an inverse Gaussian shape provides a robust approximation of both tropical and extratropical age spectra with a low amount of necessary input data. This seems a valid approach as modelled tropical age spectra (Li et al., 2012; Ploeger and Birner, 2016) exhibit strong similarities with an inverse Gaussian function only with multiple modes. Still, the constrained shape might lead to inaccuracies to an unknown extent.

Above all, the performance of the inverse method depends crucially on the physically precise quantification of the chemical lifetime for each trace gas, since chemistry and transport show a strong interrelationship. Due to this close link, the lifetime is considered to be dependent on transit times along an average Lagrangian pathway through the stratosphere (Schoeberl et al., 2000). Although Hauck et al. (2019) demonstrate that due to the consideration of multiple trace gases in the optimization process a pseudo-random error of up to ±20 % in the chemical lifetime can be compensated, the correct determination of the local lifetimes along transit time remains a considerable problem. A probable strategy could be an advancement of the ansatz by Holzer and Waugh (2015) where not only observations of (very) short-lived substances, but also a set of long-lived trace gases are involved in a Monte Carlo simulation to derive lifetimes and age spectra in stepwise fashion. The chemical lifetimes are hereby set to constant values, which describe chemistry effectively (see Sect. 2.2.4). This approach has the advantage that it reduces the influence of modelled data but relies on the goodness of the observations especially for long-lived trace gases.

### 2.2.4 Statistical inversion process

The study of Holzer and Waugh (2015) provides the basis to derive a representative chemical lifetime for each substance from observational data together with statistical techniques. For that purpose, Eq. ( 12 ) has to be slightly tweaked. All transit-time-dependent lifetimes are replaced by the concept of effective stratospheric lifetimes $\tau_{i,eff}(\vec{x}, t)$. Those quantify chemistry for any given age spectrum along all relevant transport pathways effectively by a single scalar. This leads to





$$\chi(\vec{x}, t) = \int_0^\infty \chi_{0,i}(t - t') \cdot e^{-\frac{t'}{\tau_{i,eff}(\vec{x},t)}} \cdot G_i(\vec{x}, t, t') \cdot \omega_i(t') \cdot n_i(\vec{x}, t) \cdot dt', \tag{14}$$

where $G_i(\vec{x}, t, t')$ is again parameterized by Eq. ( 3 ). The effective lifetime for any trace gas can be retrieved numerically based on a prior estimate of the age spectrum $G_i^{prior}(\vec{x}, t, t')$. If this is derived, a slightly modified version of the K-inversion algorithm will use the prior age spectrum and optimize the effective lifetime for each substance separately using the same numerical methods as above. This is the $\tau$-inversion.

The physical difference between Eq. ( 12 ) and Eq. ( 14 ) lies in the transit time gradient of the mixing ratio. Using effective lifetimes, this gradient is different from the one in the original formulation for a specified age spectrum. However, since the final mixing ratio is identical in both cases, that fact is negligible. In the original form, Holzer and Waugh (2015) apply global tropospheric lifetimes for long-lived substances to gain the prior estimate of the tropospheric age spectrum. In the stratosphere, however, most trace gases undergo considerable chemical loss processes that cannot be estimated well by global lifetimes, which make additional information necessary. Therefore, (very) long-lived trace gases are considered together with short-lived species to constrain the age spectrum. For this study, we select five short-lived brominated trace gases ($CH_2Br_2$, $CHBr_3$, $CHCl_2Br$, $CHClBr_2$, and $CH_2ClBr$), five long-lived substances ($CF_2Cl_2$ (CFC-12), $CF_2ClBr$ (Halon 1211), $CF_3Br$ (Halon 1301), $CH_3Br$, and $N_2O$), and the very long-lived trace gas $SF_6$, that has been frequently used as dynamical tracer in the past. All these species were measured during past airborne research campaigns so that a solid data basis can be established.

As stated above, the statistical inversion method requires a prior estimate of an age spectrum to infer the effective chemical lifetime. The outcome of the procedure hinges heavily on this first guess so that we introduce a Monte Carlo cross-validation (MCCV) for all tropopause sections to perturbate the dependency and also consider a variety of uncertainties. As first step, a subset of the trace gases is created, consisting of three selected species of the complete set. The subset is always composed of the dynamical tracer $SF_6$, one of the five long-lived and one of the five short-lived species. The latter two are pseudo-randomly chosen. With this subset, the first guess of the age spectrum is constructed using the $K$-inversion on Eq. ( 14 ) together with an initial guess for the effective stratospheric lifetimes of the considered species. Other than in Holzer and Waugh (2015), a global lifetime is in general not suitable, as strong local stratospheric loss processes steer the effective lifetimes along all relevant transit times for both long- and short-lived trace gases. This implies that the effective lifetime of a species is generally smaller than its global lifetime. The exception is $SF_6$, which has its main sink region in the mesosphere at large transit times. For the mainly short transport time scales in the lower stratosphere, the influence of the chemical loss is rather small, yet not negligible. The first guess effective lifetime of $SF_6$ is therefore set to be 850 years in accordance with Ray et al. (2017). In case of the short-lived substances, mixing ratios are most probably steered by local chemical loss processes around the respective entry region. First guess lifetimes for those species are taken as annual means from Tab. 1-4 in Carpenter and Reimann (2014) for the northern and tropical tropopause sections. Long-lived trace gases show the strongest difficulties when assessing the





first guess. On the one hand, global stratospheric lifetimes are likely an overestimation, as they are derived by dividing the global atmospheric burden by the global stratospheric loss rate. Local lifetimes, on the other hand, weakly regard the effective character and are in many cases derived from model simulations. This study strives for a reduced model influence, so that the global stratospheric lifetimes from Tab. 5.6 in SPARC (2013) are turned into lifetimes that consider the stratospheric burden

rather than the total atmospheric burden and treat them as first order approximation of effective loss. The effective lifetime in our formulation is similar to a transit-time-integrated steady state, but only considers trace gas burdens and sink processes above the tropopause. Since all mass above the tropopause takes only circa 10 % of the complete atmospheric mass (Volk et al., 1997), the stratospheric burden of a substance is assumed to contribute only 10 % to its global burden. Dividing burden by loss, this concept translates into lifetimes that are only 10 % of the global stratospheric lifetime. All implemented first guess

lifetimes are shown in Tab. 2.

These values provide not necessarily a valid representation so that systematic errors are included into the simulation. The errors get selected pseudo-randomly with uniform distribution for a random number of trace gases in the prior subset. They range from -50 % to +50 % for the first guess lifetimes and from $-\sigma$ to $+\sigma$ for $\chi_{0,i}(t-t')$ and $\chi(\vec{x},t)$. $\sigma$ denotes the standard deviation of a mixing ratio (see Sect. 3.2 for details). After the prior age spectrum has been determined, it is used to perform

the $\tau$-inversion on Eq. ( 14 ) during which an effective lifetime is retrieved for every remaining substance that has not been in the prior subset. Again, pseudo-random errors between $-\sigma$ and $+\sigma$ for all $\chi_{0,i}(t-t')$ and $\chi(\vec{x},t)$ are applied to this set. For solid Monte Carlo statistics, this procedure is repeated 2000 times to cover as many initial subsets as numerically feasible. There is no effective lifetime for $SF_6$, since this tracer is present in every initial subset. That is done to make full use of $SF_6$ as reasonable frame for the age spectrum, which is then convoluted with further trace gas information to get an even more robust

and unbiased prior. After completion of the Monte Carlo simulation, the median of the retrieved effective lifetimes is utilized in a final $K$-inversion for the full trace gas set to determine the desired age spectrum. $SF_6$ is also not present in this final step to keep its direct influence restricted to the prior. In the rare case that no median effective lifetime can be derived for one of the remaining substances, the species will be omitted during the final K-inversion. To estimate the uncertainty range of the simulation, the lifetime of the 25th and 75th percentile are taken to derive the upper and lower error margin of all age spectra

and related moments.

Although that procedure is numerically expensive and requires multiple simulations for one set of mixing ratios at one location, the outcome of a reduced influence of model data seems promising. This comes at the cost of relatively large uncertainties for the retrieved effective lifetimes and age spectra, which hinge strongly on precise in situ measurements. With the considered errors in the Monte Carlo simulation it is possible to receive an impression of the influence of these uncertainties on age spectra

from observations.



## 3 Data basis

### 3.1 CLaMS data

Two simulations with the Chemical Lagrangian Model of the Stratosphere (CLaMS) have been performed in a similar framework as the simulation of Ploeger and Birner (2016). CLaMS uses a Lagrangian perspective to model transport processes

and chemistry for trace gases along calculated three-dimensional forward trajectories of single air parcels in combination with a parameterization scheme for small-scale mixing (McKenna, 2002a). That scheme leads to strong mixing in regions where deformations of the background flow are large (Konopka, 2004). CLaMS simulates transport in potential temperature space, where the vertical coordinate is designed as hybrid between potential temperature in the stratosphere and upper troposphere and an orography-following pressure coordinate in vicinity to the surface with smooth transition. The vertical speed along this

coordinate is steered by the total diabatic heating rates from the reanalysis product that drives the simulation (Pommrich et al., 2014). In this study, CLaMS is driven by meteorological data of the ERA Interim reanalysis (Dee et al., 2011). The final output of the Lagrangian model is gridded spatially with a resolution of 2° by 2° and 37 vertical potential temperature levels between 280 K and 3000 K. Both simulations cover the period from January 1989 to December 2017 as daily mean. Data are evaluated as zonal and seasonal means between December 1999 and November 2009 for the proof of concept.

This setup is suitable for the lower stratosphere, since fast transport processes across the tropopause are well-resolved. Age spectra in the model are derived from completely inert trace gases, that are pulsed in certain intervals at the reference surface. Those pulse tracer series are then translated into proper spectra by the method of Ploeger and Birner (2016) for transient simulations. For convenience, these spectra are hereafter named pulse age spectra. In the first simulation (called TpSim), all tracers are initialized at the tropopause in the northern (90° N to 30° N), tropical (30° N to 30° S) and southern (30° S to 90° S)

region. Although a PV-based tropopause is a more suitable choice for dynamical studies, the simulations have been performed using the WMO definition of tropopause. For consistency between model and observations, tropopause refers hereafter always to the WMO definition. The tracer pulses are released as approximate Dirac delta distributions with a mixing ratio of one in their respective region and forced to zero when contacting the other two source sections. There are two separated sets of pulses for each source region to get a well-resolved age spectrum. They consist of 24 and 20 tracers, released monthly and

semiannually, and cover a period of two and ten years respectively. If both sets are combined afterwards, the age spectrum will provide a fine monthly resolution up to two years and a coarser semiannual resolution for the remaining transit times up to 10 years. After every tracer has been pulsed once, they are reset and re-initialized.

The simulation features also three completely inert trace gases that are constantly released with a mixing ratio of one in the three source regions. These tracers provide the origin fractions for the reference surfaces without explicit integration of the age

spectra. For the inverse method, ten trace gases with spatially constant lifetimes ranging from 1 month to 109 months in steps of 12 months are included and released globally at the reference surface. The effective and transit-time-dependent lifetimes are identical for these "radioactive" tracers. The second simulation is a copy of TpSim, but with all substances being initialized





at Earth's surface (called SurfSim) in the three specified regions. All age spectra from pulse tracers are extended to 50 years transit time using the method described in Ploeger and Birner (2016).

## 3.2 Observational data

This study uses in situ measurement data obtained during two research campaigns of the High Altitude and Long Range
Research Aircraft (HALO; www.halo.dlr.de). The first campaign, PGS (Oelhaf et al., 2019), took place during December 2015 and March 2016, with the mission base in Kiruna, Sweden. PGS was a combination of the three missions: POLSTRACC (Polar Stratosphere in a Changing Climate; www.polstracc.kit.edu), GW-LCYCLE (Gravity Wave Life Cycle) and SALSA (Seasonality of Air mass transport and origin in the Lowermost Stratosphere). PGS was split into two phases, the first from mid December 2015 till late January 2016 and the second from late February 2016 till March 2016. The focus of PGS was
strongly on the northern hemispheric upper troposphere and lower stratosphere, as well as the exchange processes around the polar vortex and arctic latitudes. The second campaign, WISE (Wave-driven Isentropic Exchange; www.wise2017.de), took place between September and October 2017, with the mission base in Shannon, Ireland. The focus of WISE was on isentropic exchange processes between troposphere and stratosphere around the mid-latitude tropopause. The flight tracks for both campaigns are shown in Keber et al. (2019). The campaign data are binned into grids of equivalent latitudes (Allen and
Nakamura, 2003) and potential temperature differences to the WMO tropopause, with a bin size of $5° \times 5$ K and treated as phase averages. Bins containing less than five data points are omitted. The standard deviation for all mixing ratios is derived during the binning procedure. All halogenated trace gases mentioned in Sect. 2.2.4 were measured by the Gas Chromatograph for Observational Studies using Tracers – Mass Spectrometer (GhOST-MS) operated aboard HALO. GhOST-MS is a dual-channel gas chromatograph coupled with an Electron Capture Detector (ECD) in an isothermal channel and a quadrupole mass
spectrometer (MS) in a temperature programed channel. The set-up and relevant precision values are given in Keber et al. (2019). All GhOST-MS data in this study are reported on SIO-05 scales. $N_2O$ was measured during PGS by the TRIHOP instrument (Schiller et al., 2008), an infrared absorption laser spectrometer with three quantum cascade lasers. The set-up for PGS and respective precision values are described in Krause et al. (2018). During WISE, $N_2O$ was measured with the UMAQS instrument (Müller et al., 2015), also an infrared quantum cascade laser spectrometer. The set-up for WISE and relevant
precisions are given in Kunkel et al. (2019). These $N_2O$ data are reported on the WMO 2006a scale.

An important parameter for the inversion is the entry mixing ratio time series at the specified tropopause sections. We only derive entry mixing ratios for the northern and tropical tropopause, as we show in Sect. 4.1 that the influence of cross-hemispheric transport is negligible. The time series should cover the period from 1960 until November 2017 to retrieve a mathematically precise age spectrum with a range of 50 years transit time. Unfortunately, there are no consistent measurements
available covering the complete period at the surface let alone at the tropopause. That is problematic for the long-lived trace gases in this study ($SF_6$, $N_2O$, CFC-12, Halon 1211, Halon 1301, and $CH_3Br$), since a strong long-term trend is detected at the surface. To construct a time series for these species, data from the Atmospheric Lifetime Experiment (ALE), the Global Atmospheric Gases Experiment (GAGE) and the Advanced Global Atmospheric Gases Experiment (AGAGE) (Prinn et al.,





2018; Prinn et al., 2019) are taken and extended backwards until 1960 with global data from the Representative Concentration Pathways (Meinshausen et al., 2011) for the two stations Ragged Point in Barbados (RPB – 13° N, 59° W) and Cape Matatula in American Samoa (SMO – 14° S, 171° W). The RCP data are aligned for any substance to suit the general behaviour of the corresponding full ALE/GAGE/AGAGE data set, but only considered where no measurements are available. All relevant data

are reported on SIO-05 scales, except $N_2O$, which is reported on both SIO-98 (ALE/GAGE) and SIO-16 (AGAGE). Despite the different scale names, $N_2O$ data on SIO-98 and SIO-16 scale are comparable. Both scales are considered to be comparable to WMO 2006A for the purpose of this study (World Meteorological Organization, 2018). Minor temporal gaps are interpolated. It is assumed that the long-lived gases are well-mixed, so that the average of RPB and SMO represents a tropical mixing ratio. As the global tropospheric lifetimes of these trace gases are sufficiently large, except for $CH_3Br$ (1.6 years

according to Tab. 5.6 in SPARC (2013)), the extended ALE/GAGE/AGAGE data is lagged to the tropical tropopause by 2 months ± 0.5 months. The validity of that approach is shown in Andrews et al. (1999) with $CO_2$ data. In case of $CH_3Br$, some chemical loss processes might already occur while propagating towards the tropopause, but since the time lag is still much smaller than the global tropospheric lifetime, the lagged mixing ratio is assumed to be representative.

For the northern hemispheric tropopause section, the net flux across the section is downward, so that tropospheric air mixes

with descending stratospheric air, characterized by lower mixing ratios due to the chemical loss regions in the stratosphere. At the extratropical tropopause the mixing ratio should thus be lower than in the tropics. Additionally, the trace gas burden at the extratropical tropopause consists of a mixture of air from the tropics and extratropics, aggravating a straightforward lag-approach. The tropical origin fraction from the CLaMS simulation SurfSim provides a suitable tool to characterize the most important surface source section. The annual mean origin fraction indicates that at the northern hemispheric tropopause section,

approximately 88 ± 4 % of all air masses originate from the surface in the tropics, with a corresponding mean AoA of 1 year ± 0.25 years. All tropical ALE/GAGE/AGAGE data is lagged by this value to retrieve the mixing ratio time series at the northern extratropical tropopause. Since the main sink of $CH_3Br$ in the troposphere is the temperature-dependent reaction with the hydroxyl radical, chemical loss processes in the cold middle and upper troposphere are again treated as first order negligible compared to the transport time scale. The standard deviation for these time series is derived from the respective

measurement error and the deviation that emerges when the uncertainty of the lag time is implemented, especially relevant for RCP data as these do not provide a measurement error. The short-lived species ($CH_2Br_2$, $CHBr_3$, $CHCl_2Br$, $CHClBr_2$, and $CH_2ClBr$) show weak long-term trends at the tropopause. For the tropics, the mixing ratios of the upper TTL in Tab. 1-4 of Engel and Rigby (2019) are considered to be the annual mean entry mixing ratio, as the potential temperature range matches the WMO tropopause of ERA Interim in the specified tropical region. The mixing ratios at the northern hemispheric tropopause

section are taken directly from the PGS and WISE data, which are averaged between 30° N and 90° N of equivalent latitude and between the tropopause and 30 K above, as this is specified as a region of strong tropospheric influence (Hoor et al., 2004). The large interval of 30 K has been introduced to incorporate the strong seasonal variability of the WMO tropopause in the northern hemisphere throughout the year into the mixing ratios and to regard general discrepancies between dynamical (PV-based) and WMO tropopause. To be consistent, the first 30 K above the tropopause are then omitted in the inversion procedure.





The uncertainty values for all short-lived substances at the northern hemispheric tropopause section are derived as the standard deviation from the average. In the tropics, the standard deviation given by Engel and Rigby (2019) is applied.

## 4 Results

### 4.1 Origin fractions for different entry regions

Origin fractions are a valuable tool to quantify the importance of air entrainment through their specified tropopause sections and cross-hemispheric transport for air mass composition in the stratosphere. Figure 2 shows global cross sections of all annually and seasonally averaged origin fractions from TpSim as function of potential temperature and latitude. The model setup seems consistent overall, as all fractions sum up to circa 100 % in the stratosphere. On annual average, the general distribution of the origin fractions resembles the pattern of the BDC quite well with strong upwelling in the tropics and a
downward motion at northern and southern extratropical latitudes. The sharp borders of all fractions at the tropopause around 30° N/S are caused by the definition of the source regions in the model. It is apparent that the tropical tropopause constitutes the predominant source region for the complete stratosphere above 450 K, with the tropical origin fraction (panel "Annual" in mid row) reaching more than 70 %. Below, the northern (panel "Annual" in top row) and southern (panel "Annual" in bottom row) tropopause sections start to gain significant influence in the extratropics manifesting in a rise of the respective origin
fractions when approaching the tropopause. Cross-hemispheric transport is negligible, since the northern fraction in the southern hemisphere and the southern fraction in the northern hemisphere are vanishingly low and only reach values up to 10 %. Therefore, also the corresponding cross-hemispheric age spectra can be omitted as they contribute only marginally to the composite spectrum. Both the northern and southern origin fractions show quite a sharp latitudinal gradient directly at the equator accompanied by a strong increase in the tropical fraction, visible as a beam of deep red shading around the equator
throughout the stratosphere. This might be an effect of the subtropical transport barriers that enclose the tropics, separate them from the extratropics and inhibit exchange processes (Neu and Plumb, 1999).

All origin fractions undergo a pronounced seasonality (panels DJF to SON in all rows). For the northern hemispheric fraction (top row), the maximum above the tropopause is visible in SON reaching up to almost 500 K in the tropics and extensive values between 50 % and 75 % in the lower stratosphere up to circa 380 K, while the minimum is found with six months offset
in MAM. JJA and DJF show a transition state between maximum and minimum, which follows the seasonality of the mass fluxes in Sect. 2.2.2. As it takes some time for the air to propagate from the northern hemispheric tropopause section upward into the stratosphere, the maximum northern origin fraction, i.e. a flushing of the northern hemisphere with fresh air, is modelled with some delay in SON. The same principle applies to the minimum of the northern fraction. Since the maximum of downward forcing through 380 K is simulated in late January, the northern origin fraction attains its minimum in MAM.
The isolated area of enhanced northern fraction at circa 380 K and 30° N in JJA could be related to the Asian summer monsoon, that is known to transport fresh air into the northern (sub)tropical stratosphere (e.g., Vogel et al., 2019). For the southern origin fraction (bottom row), the correlation with the mass flux is not as clear as in the north. Intuitively, maxima and minima should





be shifted by six months, with maximum southern fraction in MAM and minimum in SON. While the fraction in MAM appears strongest between 30° S and circa 55° S in the lower stratosphere with a pronounced vertical structure and values of 40 % to 75 %, a large area of strongly enhanced southern fraction (up to 75 %) is visible in SON at high latitudes. This is most likely linked to the initialization of the tracers at the WMO tropopause, which is found at high altitudes due to the very low

temperatures inside the southern polar vortex. Apart from that, seasonal fluctuations seem weaker in general for the southern fraction. The tropical origin fraction in the lower stratosphere shows the weakest seasonality and spreads deep into the extratropics with values around 70 % to 80 % during minimal phases of the northern and southern fraction (especially MAM and JJA in the north and JJA and SON in the south, which is in accordance with the results of Hegglin and Shepherd (2007)). The maximum of the tropical fraction follows the tropical upward mass flux (Rosenlof, 1995) and shifts from the southern

edge of the tropics in JJA to the northern edge in DJF, the latter showing a slightly broader structure.

The presented origin fractions reveal that the assumption of single entry through the TTL appears robust in the tropics and above 450 K globally. Age spectra, however, will lack important features of stratospheric transport in the extratropical lower stratosphere if only the tropical section is considered. To retrieve a precise composite age spectrum both in model and inverse method, the tropical and the respective extratropical spectrum, north or south, must be determined and superimposed. Since

the influence of cross-hemispheric transport is vanishingly low, the related cross-hemispheric age spectra are now omitted to simplify the setup.

## 4.2 Proof of concept

### 4.2.1 Age of air spectra

Hauck et al. (2019) provide an extensive proof of the inverse concept for the age of air spectra referring to the tropical

tropopause section, so this study focuses on the northern and southern spectra in the midlatitudes of the respective hemisphere close to the local tropopause. All presented age spectra are normalized to inhibit dependency on the CLaMS origin fractions. Figure 3 shows the normalized pulse (solid lines, see Sect. 3.1 for details) and inverse age spectra from the "radioactive" tracers (dashed lines, see Sect. 2.2.1 and 2.2.2 for details) with reference at the northern hemispheric tropopause at 56° N and 370 K as annual (black) and seasonal (colored) means. To ease comparison, transit times below one month are excluded from all

inverse spectra, as this is the minimum resolution the pulse tracer experiment provides. The annual mean inverse spectrum (panel (a)) matches well with the pulse spectrum and exhibits a very similar shape with one pronounced peak at similar transit times and no further modes. That indicates that also the seasonal scaling works properly as it cancels out on annual average. The amplitude of the inverse spectrum is slightly larger than the pulse spectrum, although it appears as if the mode of the pulse spectrum is clipped off by the one-month resolution leading to a slightly right-tilted peak. A more frequent pulsing of the

tracers could improve the alignment of inverse method and pulse spectra. For the seasonal spectra (panel (b) to (e)), the performance of inverse method and scaling factor seems robust with well-timed secondary minima and maxima, that agree within one month transit time between pulse and inverse spectra. The amplitude of the first mode is well reproduced in DJF and JJA, while MAM and SON are overestimated by approximately 50 %. Just as for the annual mean, the pulse spectra main





peaks in MAM, JJA, SON and slightly in DJF appear to be cut off by the resolution of the tracer pulsing with similarly right-tilted shape. Primary and secondary minima are correctly retrieved, while the corresponding secondary maxima might need a slightly stronger scaling factor than the one applied here, especially in DJF at 0.5 years and in MAM at 0.75 years. Even though the shown northern seasonal inverse spectra are independent of the respective CLaMS origin fraction, they correctly reproduce

a maximum of air entrainment in the JJA and SON spectra with almost twice as large amplitudes (4.99 a$^{-1}$ and 5.77 a$^{-1}$) as in DJF and MAM (2.12 a$^{-1}$ and 2.97 a$^{-1}$). This follows the maximum of the northern origin fraction in Fig. 2 (SON) very well and implies that the inverse spectra correctly reproduce the seasonality in cross-tropopause transport in the northern hemisphere without explicit consideration of the fraction.

Figure 4 displays correspondingly the normalized CLaMS pulse (solid lines) and inverse (dashed lines) age spectra at 56° S

and 370 K as annual (black) and seasonal average (colored) with origin at the southern hemispheric tropopause. The performance of the annual mean inverse spectrum (panel (a)) is similar as for the northern spectrum with slightly better agreement of the main mode amplitude. The timing of the peak coincides again within one month of transit time. No further modes are visible in annual mean pulse and inverse spectra indicating that also the southern hemispheric scaling factor seems to work as intended on annual scale. The fairly right-tilted shape of the pulse spectrum peak indicates as well that some features

of transport are clipped off in the spectra due to its transit time resolution. On the seasonal scale (panel (b) to (e)), the inverse method reproduces the general shape of the pulse spectra a bit better than in the northern hemisphere, with largely well-matching primary modes. Only in SON, the pulse spectra amplitude is underestimated by circa 25 %. This might be related to the WMO tropopause in this region. Consistently, all seasonal pulse spectra suffer from the same cut-off-effect at small transit times as the northern spectra. The present seasonal minima and maxima after the primary mode are robustly imposed by the

southern scaling factor with similar amplitude and match the timing of the pulse spectra within one month, except for the first minimum in SON, which agrees within 1.2 months. During southern winter (JJA), a similar underestimation of the amplitude is visible as in northern winter (DJF) of Fig. 3. This might be related to the assumed inverse Gaussian shape. In terms of transport seasonality, the inverse age spectra follow the southern origin fraction again quite well without considering them explicitly. The maximum of transport is visible in the DJF and MAM spectra with amplitudes of 4.3 a$^{-1}$ and 4.88 a$^{-1}$, while JJA

and SON constitute phases of weaker transport (2.07 a$^{-1}$ and 3.27 a$^{-1}$). Compared to the northern hemisphere, the seasonality is in general not as strong and pronounced, similar to the seasonality of the southern fraction (Fig. 2).

Despite its restriction to an intrinsically inverse Gaussian shape, the extended inverse method with the newly introduced extratropical scaling factors appears to retrieve precise age spectra in the northern and southern midlatitude lower stratosphere if chemical lifetime and mixing ratios are well-constrained. All derived inverse age spectra then capture important features of

transport from the CLaMS model on a seasonal scale without direct influence of the modelled origin fractions.

**4.2.2 Mean age of air**

For a full global scale assessment, Figure 5 shows cross sections of mean age of air derived from the composite pulse (top row) and composite inverse age spectra (bottom row). The first column (annual) shows annual mean absolute values, while the remaining four columns (DJF to SON) depict seasonal percentage differences relative to the respective annual average. A



seasonal analysis of the composite spectrum is advantageous to assess the behavior of all three different age spectra – northern, tropical and southern – simultaneously, but weighted by their geographical importance. Since all origin fractions undergo a distinct seasonality, which is not necessarily identical with the seasonality of the age spectra, the composite spectrum for this comparison is calculated using always annual mean origin fractions in Eq. ( 8 ) (inserted into Eq. ( 6 )). This ensures that the

presented seasonal pattern is only steered by the age spectra. On annual average, good agreement between inverse and pulse mean AoA is detected in general, where both show very similar spatial structures. The inverse method correctly reproduces the low mean AoA values of the pulse mean AoA in the tropics and the positive gradient towards the poles. Even the area of enhanced mean AoA at high southern latitudes between 400 K to 500 K is emulated, although it extends down below 380 K. The inverse mean AoA is generally biased and exhibits larger mean AoA than the pulse spectra. This fact is in accordance

with the results of Hauck et al. (2019), who also find an overestimation of mean AoA by the inverse method and link it to the prescribed inverse Gaussian shape of the age spectra. To quantify comparably in this study, the globally averaged bias below a threshold of 1.5 years of mean AoA and above is retrieved (see Hauck et al. (2019) for details on the threshold). We find that the deviation reduces from +44.3 % below the threshold in Hauck et al. (2019) to only +13.8 % in this study. Above, the bias remains almost steady at +12.4 % compared to +13.3 % before. This improvement demonstrates the benefit of the extended

approach, although some improvement might also be attributed to the finer pulse resolution especially around the tropopause. Seasonal differences give a similar impression with spatial patterns of inverse mean AoA that match those of the pulse mean AoA in the stratosphere qualitatively well. Only the amplitude of the differences appears enhanced for the inverse method, e.g., the darker shading of blue at 50° S and 600 K in SON but coincides with the detected bias on annual scale. In the lower stratosphere, all positive and negative fluctuations are correctly retrieved by the inverse method. That is an improvement over

Hauck et al. (2019), as they find inverted seasonal structures, i.e., positive trends in the pulse and negative in the inverse mean AoA, in the northern hemispheric lower stratosphere during MAM and SON. Only in DJF directly above the tropopause in the northern midlatitudes and at the south pole the shadings appear different. That might be an artifact of the close proximity to the tropopause where an inverse Gaussian shape might not resemble the pulse spectrum correctly. Both pulse and inverse mean AoA exhibit a flushing of the sub- and extratropical lower stratosphere with fresh air during summer and fall of the respective

hemisphere. That coincides well with the season of maximum amplitude of the northern and southern age spectra in Fig. 3 and 4 and shows their importance for a seasonally precise description of transport in the lower extratropical stratosphere. In the tropics, the maximum of tropospheric air is visible in MAM, but some strong entrainment is already visible around 30° N in DJF and around -30° S in JJA. This follows the seasonality of the tropical origin fraction in the tropics shown in Fig. 2 without explicit inclusion of the seasonal factors into the composite age spectrum.

The results of the idealized proof of concept demonstrate the significantly improved performance of the extended inverse ansatz for age spectra in the lower extratropical stratosphere, which has previously been identified as critical region for the tropical age spectra by Hauck et al. (2019). The inverse method retrieves the northern and southern age spectra correctly and the newly inferred seasonal cycles impose modes at transit times that correspond to the CLaMS pulse age spectra very well both locally in the northern and southern midlatitudes and on the global scale as composite with the tropical spectra. In its





extended state, the inverse method can probably provide insight into transport mechanisms involving the tropical and extratropical tropopause. However, since this section provided only a highly idealized test scenario, the performance of the method and the statistical retrieval procedure for chemical lifetimes is assessed under more realistic conditions in the next sections.

## 4.3 Observational data

### 4.3.1 Mean age of air

The focus of the following sections is on the application of the inverse method on observational data. Results are evaluated under consideration of findings in previous studies. Note that the following sections use solely equivalent latitude as horizontal and potential temperature difference to the local tropopause as vertical coordinate. All presented age spectra are independent of any modelled origin fractions. Figure 6 depicts cross sections of mean AoA from the normalized inverse age spectra referring to the northern hemispheric (top row) and tropical tropopause (bottom row) during PGS phase 1 (first column), PGS phase 2 (second column) and WISE (third column). The spatial distribution and quantitative range of inverse mean AoA in both rows appears meaningful and coherent in general showing smaller values towards the tropics and an increase with latitude and altitude. For PGS phase 1 and PGS phase 2, the spatial distribution seems consistent with the data in Fig. 3 of Krause et al. (2018), although their observational-based mean AoA values refer to Earth's surface in the tropics and therefore regard transport across both tropical and northern hemispheric tropopause. The quantitative range of mean AoA in Krause et al. (2018) should be larger than in this study, as tropospheric transport up to the tropopause sections is included into their mean AoA.

In case of mean AoA with origin at the northern hemispheric tropopause (top row) it is found that with the applied inverse method setup, PGS phase 2 displays the largest mean AoA of all data with scattered bins of more than 3 years mean AoA around 90 K and 75° N. While both PGS phase 1 and PGS phase 2 cover a wide latitudinal range from 35° N up to 85° N, WISE is strongly confined and centered around 50° N with similar vertical extent as PGS 1. For WISE, the inverse method derives mean AoA that is slightly smaller than mean AoA during PGS in the same spatial region. Minimum inverse mean AoA values of all three campaigns are retrieved for PGS phase 1 between 40° N and 45° N below 50 K and even below 40 K at circa 70° N with bins of less than 0.1 years. However, mean AoA during WISE in the same spatial region might be of equal size if data were present. These findings imply a strong entrainment of fresh tropospheric air into the lowermost stratosphere across the northern hemispheric tropopause during fall. On the one hand, this manifests in already diminished mean AoA during WISE, i.e., early fall 2017, and, on the other hand, in minimum mean AoA values for PGS phase 1, i.e., winter 2015/2016, where air that entered prior to the campaign had already some time to propagate upward from the tropopause. That seasonality in local entrainment across the tropopause is consistent with the results of the SPURT aircraft campaign in Fig. 6 of Bönisch et al. (2009), showing a maximum of air with tropospheric origin in the lowermost stratosphere in October (> 80 %) and a minimum vice versa in April (< 20 %) due to strong local quasi-isentropic mixing processes.



For the inverse age spectra that refer to the tropical tropopause (bottom row), it is evident that derived mean AoA is larger than the northern counterpart, with the average difference ranging from +0.3 years for WISE to +0.5 years for PGS phase 1 and phase 2. Maximum mean AoA is retrieved for PGS phase 2 with values of more than 4 years but with a larger vertical extent down to 55 K at 75° N. Similar as above, minimum mean AoA in the midlatitudes between 50° N and 70° N is found during

WISE, but only slightly smaller than during PGS. The absolute minimum of mean AoA during all campaigns is retrieved again in PGS 1 (~0.1 years at 40° N and 50 K), although the spatial distribution of the minimum is much more confined to low latitudes than for mean AoA with the northern hemispheric tropopause as reference. The generally lower mean AoA values derived during WISE with origin at the tropical tropopause is expected, as northern hemispheric winter is characterized as season where the tropical upward mass flux attains its maximum (Rosenlof, 1995). Therefore, entry of fresh tropospheric air

through the tropical tropopause peaks during northern hemispheric winter and manifests in lower mean AoA with some delay during JJA and SON in the northern extratropical lowermost stratosphere.

Although these findings coincide robustly with results of previous studies, the strong similarity between the spatial distribution of mean AoA with northern and tropical tropopause as reference is quite unintuitive. Since transport processes to a specified location starting at the northern extratropical tropopause should be different from that beginning at the tropical tropopause,

one could expect that mean AoA fields are more individually shaped. To check that this is not caused by the inversion concept in general, the raw model data of CLaMS TpSim (see Sect. 3.1) have been interpolated onto the HALO flight tracks for PGS and WISE. CLaMS pulse mean AoA fields are shown together with the corresponding inverse mean AoA in Fig. S1 and S2 in the supplement to this study. Results reveal that CLaMS models a similarity between mean AoA with origin at the northern hemispheric and tropical tropopause analogous to the inverse method based on observational data.

**4.3.2 Campaign-averaged age of air spectra and mean age of air**

Figure 7 presents the campaign-averaged age spectra derived by the inverse method with reference at the northern hemispheric tropopause (panel (a)) and tropical tropopause (panel (b)) for PGS phase 1 (blue), PGS phase 2 (green) and WISE (orange). To ensure comparability, the campaign average is constructed by selecting only datapoints that are finite in both PGS phases and WISE. Shaded areas denote the derived uncertainty range from the Monte Carlo simulation (see Sect. 2.2.4 for details).

For the inverse spectra with reference at the northern hemispheric tropopause the maximum amplitude is detected during WISE (9.49 $a^{-1}$), followed by PGS phase 1 (5.83 $a^{-1}$) and PGS phase 2 (1.13 $a^{-1}$). The transit times at the spectra maxima (i.e, modal age) come out equally for WISE and PGS phase 1, both around 0.5 months. This implies that a flushing event with extratropical tropospheric air due to mixing across the northern tropopause section is retrieved for early fall 2017 prior to WISE and early winter 2015/2016 prior to PGS phase 1. That is corroborated by the inverse spectrum for PGS phase 2, that displays its first

mode at a modal age of circa two months, equivalent to mid-winter 2015/2016. While the age spectrum for WISE is rapidly decreasing after its primary mode and reaches its first minimum at circa 0.8 years transit time, the inverse age spectra during PGS phase 1 and 2 exhibit a saddle point up to transit times of 0.5 years and 0.75 years respectively. However, those secondary peaks are parametrized by the seasonal scaling factor and can therefore not be considered as real signal of transport. Mean



AoA values for the spectra in panel (a) of Fig. 7 are shown in panel (a) of Fig. 8 and quantitatively emphasize the seasonality in transport visible on a larger scale in Fig. 6. For WISE, mean AoA is retrieved to be considerably lower (0.24 years) than for PGS phase 1 (0.67 years) and phase 2 (1.10 years), which is in accordance with the seasonality found by Bönisch et al. (2009) for the SPURT campaign.

All primary modes of the age spectra with origin at the tropical tropopause show smaller amplitudes and generally broader peaks compared to the northern counterparts but with an identical order of the campaigns. The maximum is found for WISE (5.92 a$^{-1}$), followed by PGS phase 1 (2.79 a$^{-1}$) and PGS phase 2 (1.06 a$^{-1}$). Modal ages are similar for WISE and PGS phase 1, both with circa one month. For PGS phase 2 an increase is visible reaching a modal age of three months. These age spectra imply that entry of fresh tropospheric air through the tropical tropopause has peaked in early fall 2017 and also early winter

2015/2016, but less strong than for the northern inverse spectra. This a rather unexpected feature, since according to the seasonality in the tropical upward mass flux in northern winter, the maximum of the age spectra with reference at the tropical tropopause should be located at transit times that correspond to winter (e.g., 0.75 years for a spectrum in SON). A possible reason might be the shallow branch of the BDC in close proximity to the tropopause. Air that enters through the tropical tropopause throughout the year is then rapidly conveyed to the lowermost extratropical stratosphere and masks the seasonality

of the tropical upward mass flux. Corresponding campaign-averaged mean AoA values in panel (b) of Fig. 8 match the general tendency of mean AoA in the bottom row of Fig. 6. Lowest values are retrieved for WISE (0.51 years), while mean AoA of PGS phase 1 and phase 2 shows larger values (1.16 years and 1.48 years respectively).

To check again that these features are not caused by the inversion procedure, campaign-averaged pulse age spectra interpolated from CLaMS TpSim are shown together with the retrieved inverse age spectra in Fig. S3 in the supplement. It shows that

CLaMS models similar age spectra for PGS and WISE as retrieved by the inverse method without direct influence of model data on the inversion.

### 4.3.3 Campaign-averaged ratio of moments

Multiple studies in the past focused on the derivation of age of air spectra and mean AoA from observations in the lower stratosphere and constrain not solely the shape of the age spectrum by the inverse Gaussian function of Eq. ( 3 ) but also regard

a constant ratio of variance to mean AoA. This quantity is called ratio of moments $\mu$. For instance, Volk et al. (1997) consider a ratio of moments of 1.25 ± 0.5 years between 60° N and 70° S up to 20 km altitude, while Engel et al. (2017) and previous assessments use 0.7 years for the northern hemispheric midlatitudes up to 30 km. Those values are based on model results by Hall and Plumb (1994) and might be an underestimation, since Hauck et al. (2019) demonstrate in their model simulation that the tail of the spectrum amplifies the ratio of moments considerably. They propose a ratio of moments of two years in the

midlatitude lower stratosphere on annual average, but state that a seasonality in $\mu$ is present. Recently, the significant influence of the ratio of moments on the derivation of mean AoA from SF$_6$ measurements is further evaluated by Fritsch et al. (2019). This study provides a suitable frame to re-assess the assumptions for the ratio of moments. Therefore, the campaign-averaged ratios of moments for the inverse age spectra in Fig. 7 are displayed in right panel of Fig. 8. It is evident that the ratio undergoes





a seasonality for the inverse spectra referring to the northern hemispheric tropopause section (panel **(c)**) and the maximum and minimum are retrieved as 1.21 years in PGS phase 1 and as 0.52 years during WISE respectively. The seasonality therefore differs slightly from that found for mean AoA. The shown quantitative range of the retrieved ratio of moments matches the applied values of Volk et al. (1997) and Engel et al. (2002) reasonably well, although a solely constant value might not fully

capture seasonal variations of the age spectra. In case of the inverse age spectra with reference to the tropical tropopause (panel (d)), values for the ratio of moments are found to range from 1.08 years minimum in WISE up to 2.81 years maximum in PGS phase 1. The seasonality pattern is again different from the corresponding mean AoA seasonality, but similar to the ratio of moments derived from the age spectra with northern hemispheric tropopause origin. PGS phase 1 for the tropical tropopause age spectra is the only data set where the ratio of moments can neither be found in the range of values used by Volk et al.

(1997) and Engel et al. (2002).

In order to constrain matching seasonal age spectra when using ratio of moments, a value of 0.7 years or $1.25 \pm 0.5$ years might not be matching universally, since results for PGS and WISE reveal that $\mu$ succumbs a pronounced seasonality. The inverse method could be considered as promising alternative as the presented results show a robust performance for northern and tropical age spectra and mean AoA on a seasonal and wider geographical scale without prior constraints to the moments of the

spectra. Especially as the presented results demonstrated the good performance of the method compared to previous studies on transport in the northern hemispheric lowermost stratosphere. However, the statistical uncertainties remain considerably high. The following section therefore provides a summary of the results as well as a critical discussion of the methods capabilities and its limitations.

## 5 Summary and discussion

This study presents an extension and application of the inverse method by Hauck et al. (2019) to derive age spectra from trace gas mixing ratios in the lowermost stratosphere by considering entry of tropospheric air across a northern (90° N – 30° N), a tropical (30° N – 30° S) and a southern (30° S – 90° S) tropopause section. The age spectrum shape is predefined as inverse Gaussian function (Hall and Plumb, 1994), but with multiple modes that are imposed by seasonal parametrizations and scale the age spectrum during the inversion according to the seasonality of the stratospheric mass flux. In the first part of this study,

the concept of the extended method is tested in a CLaMS model simulation framework using ten artificial trace gases with globally constant chemical lifetimes for the inversion. Resulting annual and seasonal mean inverse spectra are compared to CLaMS pulse age spectra as reference. The simulation additionally features origin fractions that quantify the percentage fraction of air at a stratospheric point that entered across the specified northern, tropical and southern tropopause region. In the second part, the extended inverse method is applied to observational data of short- and long-lived halogenated trace gases

gained in the northern hemisphere lower stratosphere during the research campaigns PGS  and of the HALO research aircraft. A Monte Carlo cross-validation is introduced to retrieve age spectra and chemical lifetimes for the considered species in



stepwise fashion taking the variability of mixing ratios and uncertainty of lifetimes into account. Derived inverse age spectra, mean AoA and ratio of moments are assessed under consideration of results in previous studies.

The newly established origin fractions turn out to be a valuable tool to assess and quantify the importance of different regions for cross-tropopause transport on a seasonal scale. Model results indicate solidly that above 450 K the stratosphere is

prevalently steered by entrainment across the tropical tropopause throughout the year. Below, transport across the northern and southern tropopause gains influence, but only for the related hemisphere as cross-hemispheric transport processes appear negligible in all seasons in the model. The maximum of entrainment across the northern tropopause section is found in general during SON. That coincides with the findings Bönisch et al. (2009), which show an enhancement of quasi-isentropic mixing across the weak northern hemispheric subtropical jet stream during summer and fall. In the south, a weaker seasonality with a

more complex spatial structure is found. The maximum of intrusion in the southern midlatitudes can be detected in MAM in agreement with the northern hemispheric seasonality. These seasonality patterns are in accordance with the findings of multiple studies of seasonality in troposphere-stratosphere exchange (Appenzeller et al., 1996; Olsen et al., 2004; Škerlak et al., 2014). The performance of the inverse method in the idealized proof of concept seems consistent and retrieval of respective age spectra in northern and southern midlatitudes at 370 K works soundly. Cross-hemispheric age spectra are omitted due to the

negligible fractions. The decoupled inversion for the three spectra works properly and the parametrized seasonal cycles impose multiple modes at congruent transit times (within 1.5 months). The derived amplitude is similar to the corresponding pulse spectra reference, although the general shape of the inverse peaks appears smoothed. Some discrepancies between inverse and pulse spectra occur around the first mode (e.g., SON in the northern and DJF in the southern spectra). These are likely a consequence of the one-month transit time resolution that clips off the peak in the pulse spectra for very short transit times.

For a global scale comparison, mean AoA from the composite age spectra (sum of all relevant sub-spectra) is considered, but derived using annual mean origin fractions to examine solely the seasonality in mean AoA. The general agreement of inverse and pulse mean AoA proves to be robust matching both spatially and quantitatively on annual average. The positive bias between inverse and pulse spectra decreases significantly by 30.5 % below 1.5 years mean AoA and by 3.5 % above 1.5 years mean AoA compared to the values in Hauck et al. (2019). The seasonal cycle of mean AoA is correctly reproduced during all

seasons although a slight overestimation of the amplitude by the inverse method is visible. Especially in the northern lowermost stratosphere in MAM and SON the seasonality is now correctly reproduced by the inverse method in contrast to Hauck et al. (2019). The improved performance of the inversion compared to the previous study is apparent and indicates the importance of transport across the extratropical tropopause for correctly retrieved seasonal age spectra in the vicinity to the tropopause. Admittedly, some improvements are certainly attributed to the fine transit time resolution (one month) of the pulse spectra in

CLaMS. If the resolution is increased in future simulations, the agreement of spectra and mean AoA will probably further advance as well, due to fully captured first modes in the age spectra.

For PGS and WISE data, the inverse method retrieves age spectra and mean AoA with meaningful quantitative range, spatial and seasonal features for both the tropical and northern hemispheric tropopause region. The derived spatial distribution of mean AoA for PGS phase 1 and 2 coincides well with the results by Krause et al. (2018), although their presented mean AoA



from $SF_6$ measurements is referred to the surface in the tropics and therefore exhibits generally larger values. Retrieved mean AoA referring to the northern hemispheric tropopause is lower during WISE than during PGS phase 1 and phase 2. This seasonal feature coincides well with the findings of Bönisch et al. (2009) for a different aircraft campaign, who found the tropospheric influence in the northern extratropical lowermost stratosphere maximal during October and minimal during April.

This could be an indication for strong quasi-horizontal mixing processes across the extratropical tropopause in the north prior to WISE. Campaign-averaged inverse age spectra display consistently a strong entrainment of tropospheric air across the northern tropopause section in fall 2017 and also with reduced strength in early winter 2015/2016. Mean AoA with reference at the tropical tropopause section is found to be larger than northern mean AoA, with an average difference of +0.3 years for WISE and +0.5 years for PGS phase 1 and phase 2. As for the northern tropopause section, mean AoA with origin at the

tropical tropopause shows lowest values for WISE and again an increase between PGS phase 1 and phase 2. This is consistent with the seasonality in the tropical upward mass flux having a maximum during DJF (Rosenlof, 1995) accompanied by a decrease of mean AoA with some delay in summer and fall (i.e., WISE). Campaign-averaged inverse spectra indicate a strong intrusion across the tropical tropopause prior to WISE and PGS. This unexpected feature might be linked to the shallow branch of the BDC that conveys freshly entered tropospheric air all-year from the tropics to the extratropical lowermost stratosphere

in vicinity to the tropopause and interferes with the seasonality in the tropical upward mass flux. To verify that the strong similarity between mean AoA and campaign-averaged age spectra for the northern hemispheric and tropical tropopause section is not artificially caused by the inversion, they are compared to data of CLaMS TpSim interpolated onto HALO flight tracks. Results reveal that CLaMS models an alike similarity as the inverse method without consideration of CLaMS data in the inversion. For a thorough assessment, the ratio of moments is presented for all campaigns, being an important quantity for the

derivation of mean AoA from $SF_6$ and $CO_2$ in the past (Volk et al., 1997; Engel et al., 2002). Previous studies assume a constant ratio of moments, usually between 0.7 years and 1.75 years, for many spatial regions in the stratosphere. Campaign-averaged results of the inverse spectra in this study indicate that the ratio of moments succumbs a significant seasonality, ranging from 0.52 years (WISE) to 1.21 years (PGS phase 1) for the age spectra with northern tropopause reference and from 1.08 years (WISE) to 2.81 years (PGS phase 1) for age spectra with tropical tropopause reference. This seasonality could be incorporated

in future studies for a precise mean AoA retrieval when using the ratio of moments to constrain the age spectrum.

Although the presented results show a robust performance of the inverse method for the application to observational data, where seasonal and structural key features of transport are well emulated and congruent with findings of earlier studies, there are multiple critical aspects that must be recognized. Although inverted age spectra and related moments retrieved from PGS and WISE data are compared to some findings in previous studies, a thorough comparison is difficult as past studies use

different reference surfaces as in this study. As comparable observationally derived mean AoA values and age spectra could become available in the future, a proper comparison with the inverse method is an important task for future studies. Moreover, inverse age spectra are restricted to the seasonal scale and an extension to finer scales (monthly etc.) might be useful to incorporate rapid transport processes but remains difficult due to increasing variability. As indicated, the overall uncertainty of the inverse spectra and their moments is very large and multiple factors contribute to that highly uncertain nature.





The most critical aspect are the derived effective lifetimes for the species considered in this study. Holzer and Waugh (2015) indicate that this concept is applicable to derive transit time spectra in the troposphere, but errors grow significantly for stratospheric application due to the strong chemical loss process and spatial variability of most of the trace gases. We quantify these uncertainties partly by the Monte Carlo simulation to examine a variety of initial sets of trace gases together with strong

statistical errors, but it is not feasible to include all possible states in the inverse method. That implies that the provided error range should be treated as a minimum. Effective lifetimes must be considered as highly theoretical concept and cannot be interpreted without their associated age spectrum and mixing ratio of the trace gas. A comparison to known global or local stratospheric lifetimes is not useful, since effective lifetimes describe chemistry along a pathway determined by the underlying age spectrum. For completeness, resulting effective lifetimes are shown without further discussion in Fig. S4 of the supplement

to this study for the campaign-averaged inverse age spectra (Sect. 4.3.2). Future studies could re-assess our results by using modelled chemical lifetimes that depend explicitly on transit time from a pulsing experiment similar to Plumb et al. (1999), but resulting age spectra will then depend strongly on the chosen model setup.

This study tries to achieve a reduction of model influence by separate consideration of tropical and northern age spectra, although some information from global atmospheric models is inevitably necessary. For a fully retrieved composite age

spectrum the origin fractions must be provided by a model. While the entry mixing ratio time series for all long-lived species are primarily constructed from ALE/GAGE/AGAGE measurements and extension back to 1960 using aligned mixing ratios from the RCP data set, which might not certainly constitute a precise description. Also, the time lag from the surface to the tropopause, which is assumed to be constant, might cause inaccuracies, especially in case of the northern tropopause, since the time lag is directly taken from a CLaMS simulation. All these uncertainties are considered within the applied error of entry

mixing ratios of the Monte Carlo simulation, but it is not provided that they are captured to full extent. Improvements in measurement networks and technologies in the future could provide more accurate data for the tropopause sections and lead to improved age spectra. The same applies to the measured mixing ratios during PGS and WISE. These are always processed together with their standard deviation, but variability and data quality in general is a crucial factor, that influences the inverted age spectra significantly and contributes to the large uncertainty range of the results. Improvements to measurement data in

the future, even for single species, could lead to an enhanced performance. Finally, the inverted age spectra must be evaluated carefully if assessing seasonality in transport. While seasonal shifts in mean AoA and age spectra in general are resolved via changes in the mixing ratios, variability of stratospheric mass flux, which causes the multimodality of the age spectra, is not included in the method. This is due to higher order peaks being imposed with a fixed factor at predefined transit times. That inhibits a seasonal analysis involving higher modes of the inverse spectra. Nevertheless, the results demonstrate that with the

improvements to the inverse method in this study, age spectra an mean AoA can be inferred from mixing ratio measurements of atmospheric trace gases and deliver plausible results for seasonal aspects of stratospheric transport in the northern hemisphere lower stratosphere. Although results must always be seen in the light of their uncertainty range, additional information on top of mean AoA can be retrieved by inclusion of further chemically active trace gas species. This might contribute to a deepened understanding of seasonal variability for future studies.





**Data Availability**

In-situ data from HALO are available via the HALO database (http://halo-db.pa.op.dlr.de). ALE/GAGE/AGAGE and RCP mixing ratio data are available online (http://agage.mit.edu and http://www.pik-potsdam.de/~mmalte/rcps/). CLaMS model data can be made accessible on request to the authors.

**Competing interests**

The authors hereby declare that they do not have conflicting interests.

**Author contribution**

MH wrote the manuscript, performed the data processing and evaluation and prepared the figures for this paper. MH and AE developed the extended principle of the inverse method in close collaboration. FP and MH planned, conducted and
postprocessed the CLaMS simulations. AE, HB, PH, TK and FP were an active part in the PGS campaign and in the evaluation of data. AE, MH, PH, TK, FP and TJS were an active part in the WISE campaign and in the evaluation of data. All co-authors contributed to the research in this paper during many discussions.

**Acknowledgements**

This work is funded by the German Research Foundation (DFG) priority program 1294 (HALO) under the project number
316588118. The model simulations were partly performed on the HPC system JURECA of the Jülich Supercomputing Centre (JSC) at Forschungszentrum Jülich. The authors greatly acknowledge all people involved with the ALE/GAGE/AGAGE network for their persistent effort in measuring atmospheric gas constituents and for provision of their data. Especially people in charge at the ALE/GAGE/AGAGE stations American Samoa (J. Mühle, C. Harth, P. B. Krummel and R. Wang) and Ragged Point (S. O'Doherty, D. Young, P. B. Krummel and R. Wang). AGAGE is supported principally by NASA (USA) grants to
MIT and SIO, and also by: BEIS (UK) and NOAA (USA) grants to Bristol University; CSIRO and BoM (Australia): FOEN grants to Empa (Switzerland); NILU (Norway); SNU (Korea); CMA (China); NIES (Japan); and Urbino University (Italy). Finally, the authors like to thank all organizers and participants of the PGS and WISE campaign, which provided the frame to retrieve the necessary data for this study.

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

**Figure 1:** Seasonal scaling factors for age spectra referring to the northern hemispheric tropopause section (left – **(a)**) and southern hemispheric tropopause section (right – **(b)**). Month-based abbreviations are used so that identical seasons have inverted colors in both hemispheres.







**Figure 2:** CLaMS origin fractions from TpSim as annual (left column) and seasonal (remaining four columns) mean global cross sections from the northern hemispheric (top row), the tropical (mid row) and the southern hemispheric (bottom row) tropopause source region (for definition see text). The solid white line indicates the WMO tropopause from ERA Interim data. Negative latitudes always denote the southern hemisphere.



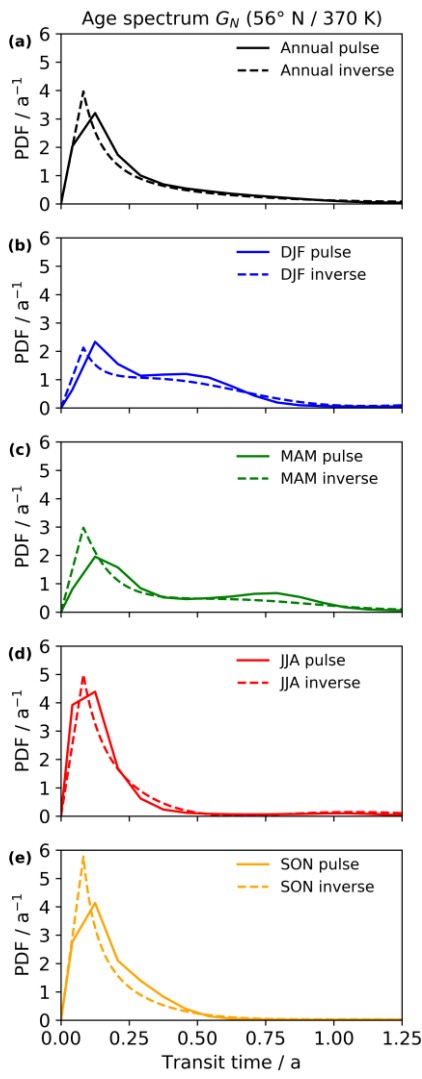

**Figure 3:** Normalized age spectra with reference at the northern hemispheric tropopause region $G_N$ at 56° N and 370 K as annual (panel **(a)**) and seasonal (panel **(b)** – **(e)**) means. Solid line denotes age spectra from CLaMS pulse tracers, dashed line inverse method age spectra.



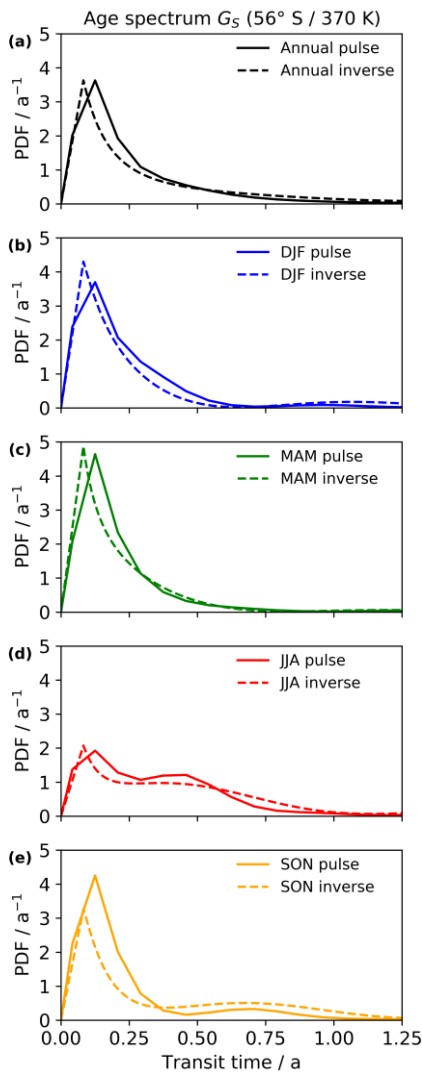

**Figure 4:** Normalized age spectra with reference at the southern hemispheric tropopause region $G_S$ at 56° S and 370 K as annual (panel **(a)**) and seasonal (panel **(b)** – **(e)**) means. Solid line denotes age spectra from CLaMS pulse tracers, dashed line inverse method age spectra.

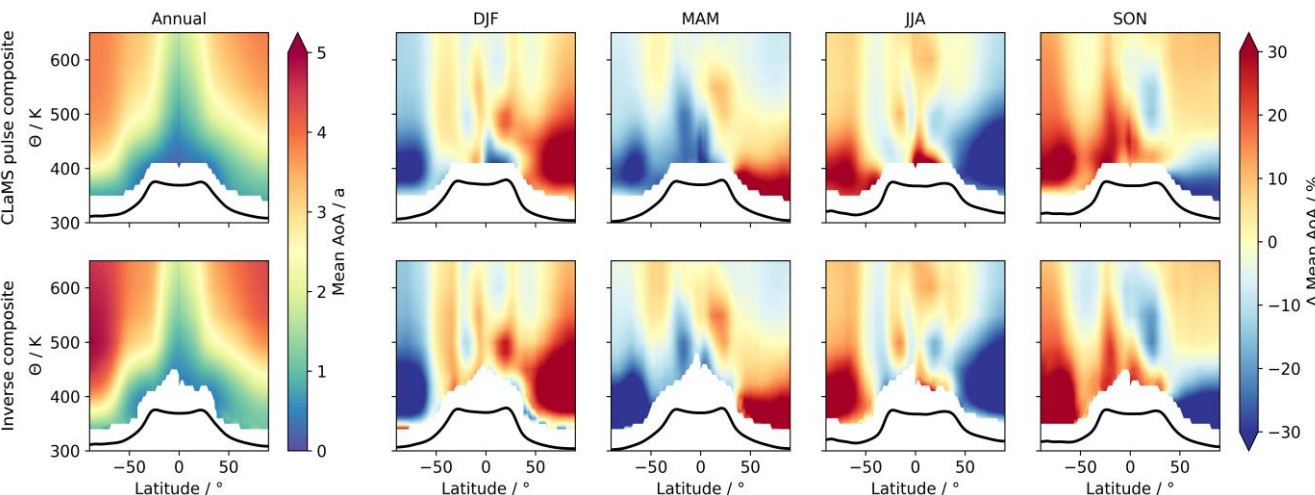

**Figure 5:** Global cross sections of mean age of air derived from the CLaMS pulse composite age spectra (top row) and from the inverse composite spectra (bottom row). First column shows absolute values as annual mean and the right four columns seasonal percentage differences relative to the annual mean. In all panels, the black line indicates the tropopause. The composite age spectra are calculated using only annual mean origin fractions of the CLaMS model to focus explicitly on seasonality in mean AoA. Note that in the bottom row the larger areas of undefined values at the tropical tropopause are caused by the inversion algorithm not finding a valid solution for the transport parameter in that region. The first 30 K above the tropopause are omitted in all panels.





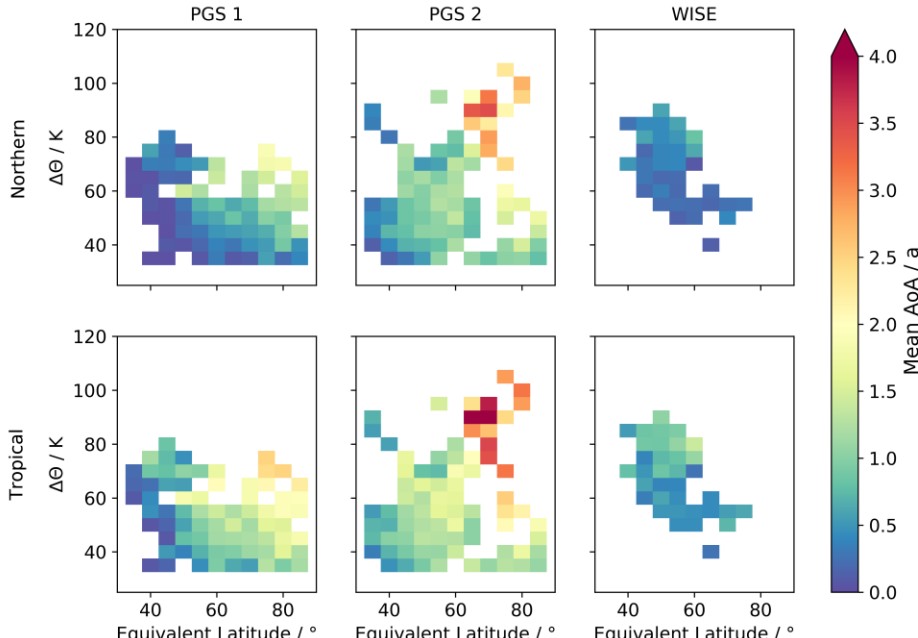

**Figure 6:** Cross sections of binned mean age of air calculated using the age spectra retrieved with the inverse method in the Monte Carlo simulation (see Sect. 2.2.4). Top row shows mean AoA referring to the northern hemispheric tropopause section, while mean AoA in the bottom row refers to the tropical tropopause section. Left column displays data of PGS phase 1, mid column data of PGS phase 2 and right column data of WISE. The potential temperature difference to the local tropopause ΔΘ is used as vertical coordinate, equivalent latitude as horizontal coordinate.



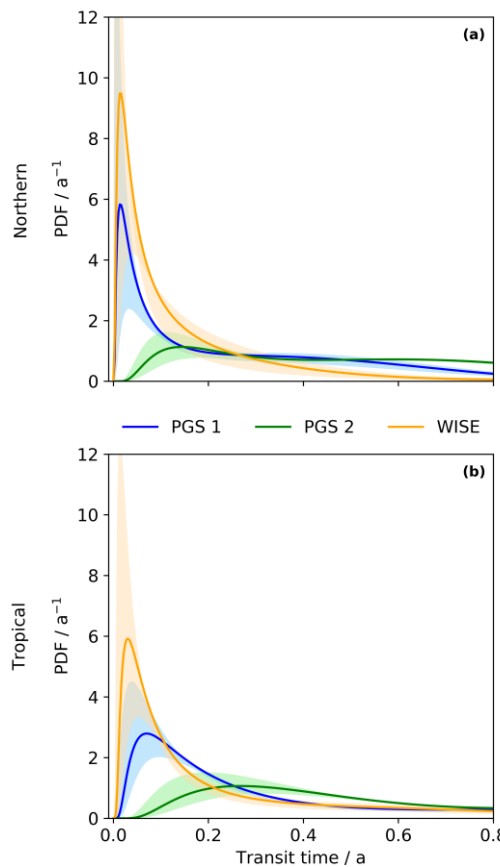

**Figure 7:** Campaign-averaged age spectra for PGS phase 1 (blue), PGS phase 2 (green) and WISE (orange). Panel **(a)** shows normalized inverse age spectra $G_N$ referring to the northern hemispheric tropopause section, while panel **(b)** shows normalized inverse age spectra $G_T$ referring to the tropical tropopause section. Colored shadings give the uncertainty range of the spectra derived from the Monte Carlo simulation (see Sect. 2.2.4 for details).





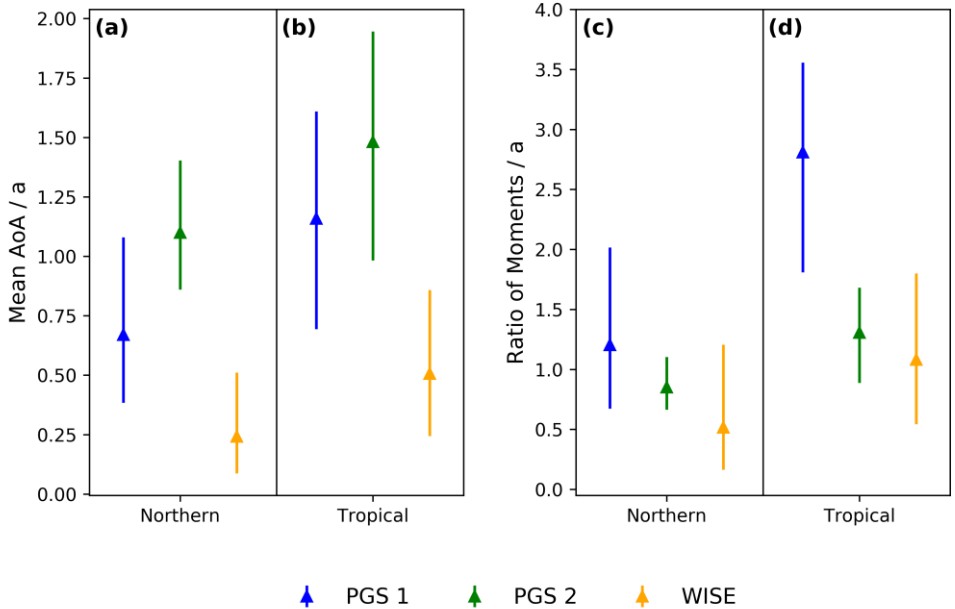

**Figure 8:** Campaign-averaged mean AoA (panel **(a)** and **(b)**) and ratio of moments (panel **(c)** and **(d)**) derived from inverse age spectra for PGS phase 1 (blue), phase 2 (green) and WISE (orange). Data in panels **(a)** and **(c)** refer to the northern hemispheric tropopause section, while data in panels **(b)** and **(d)** use the tropical tropopause as reference. Error bars denote the mean uncertainties from the Monte Carlo simulation (see Sect. 2.2.4).





**Table 1:** Scaling constants of Eq. ( 13 ) for the northern, tropical and southern tropopause section $(N, T, S)$ in all four seasons. The tropical values are taken from Hauck et al. (2019).

|  | $A_T$ | $B_T$ | $C_T$ | $A_N$ | $B_N$ | $C_N$ | $A_S$ | $B_S$ | $C_S$ |
|---|---|---|---|---|---|---|---|---|---|
| **DJF** | 0.8 | 0.2 | $0.0 \cdot \pi$ | 2.00 | $-1.166$ | $-0.172 \cdot \pi$ | 0.65 | 0.583 | $-0.295 \cdot \pi$ |
| **MAM** | 1.0 | 0.25 | $1.5 \cdot \pi$ | 0.90 | 0.316 | $0.398 \cdot \pi$ | 1.45 | $-1.142$ | $0.371 \cdot \pi$ |
| **JJA** | 1.3 | 0.3 | $1.0 \cdot \pi$ | 0.65 | 0.583 | $-0.295 \cdot \pi$ | 1.85 | $-1.134$ | $-0.230 \cdot \pi$ |
| **SON** | 1.0 | 0.25 | $0.5 \cdot \pi$ | 1.15 | -0.474 | $0.398 \cdot \pi$ | 0.80 | 0.473 | $0.361 \cdot \pi$ |

5  **Table 2:** Initial guess effective lifetimes for all species in this study. The values are annual averages and taken from Ray et al. (2017), SPARC (2013) and Carpenter and Reimann (2014).

|  | **Northern & Southern** | **Tropics** |
|---|---|---|
| $SF_6$ | 850 a | 850 a |
| $N_2O$ | 11.6 a | 11.6 a |
| **CFC-12** | 9.6 a | 9.6 a |
| **Halon 1211** | 3.4 a | 3.4 a |
| **Halon 1301** | 7.4 a | 7.4 a |
| $CH_3Br$ | 2.6 a | 2.6 a |
| $CHBr_3$ | 45 d | 17 d |
| $CH_2Br_2$ | 451 d | 150 d |
| $CHCl_2Br$ | 128 d | 48 d |
| $CHClBr_2$ | 89 d | 28 d |
| $CH_2ClBr$ | 529 d | 174 d |