# Peer review of "A convolution of observational and model data to estimate age of air spectra in the northern hemispheric lower stratosphere"

_Atmospheric Chemistry and Physics, 2020_

## Referee Comment (RC1) · Anonymous Referee #3 · 20 Apr 2020

This study presents an extension of the inverse methodology in Hauck et al. (2019) to derive stratospheric age of air from mixing ratios of a set of tracers, including entry of air masses through the tropical and extratropical tropopause of both hemispheres with corresponding seasonality scaling factors. The methodology is in general valid and a novel result is an important role of upward transport from the extratropical tropopause, which could help explain inconsistencies in previous age of air results in the lowermost stratosphere. The paper is well written, especially the methodology sections. However, some clarification is needed on the proposed processes behind the results before I can recommend publication, in particular there is some confusion regarding the seasonality of the mass flux from the extratropical tropopause.

[Figure]

General comments

1) The discussion of the seasonality of upward flux from the extratropical tropopause and the obtention of scaling factors is quite confusing. In Section 2.2.2 (Extratropical seasonal cycles) it is stated that the scaling factor obtained based on previous works implies maximum upward flux from the tropopause into the stratosphere in spring, and minimum in fall (lines 23-26, ll27-32 page 8). In contrast, the paper results suggest the opposite seasonality, with maximum in SON for the NH (e.g. lines 7-8 page 23). Nevertheless, the authors state that their results agree with previous works (e.g. lines11-12 page 23).

Importantly, I believe there is a wrong interpretation of the results in Fig. 6 of Appenzeller et al. (1996), which show the 'net mass flux across the tropopause due to mass variation of the lowermost stratosphere alone', that is, the dM/dt term in their Eq. 1. This flux is considered here as 'the net flux across the tropopause', which is then argued to change sign with season (P8L4-5). However, the net flux across the extratropical tropopause is shown in Fig. 8 of A96, and corresponds with the term Fout in their Eq. 1. This flux is downward (negative) year-long, as argued also by subsequent works (including Olsen et al. 2004 cited here, see their Figure 2).

The seasonal cycle of the scaling factors is obtained taking reciprocal values of the 380 K downward flux from Olsen et al. (2004). This is justified by saying that "this downward motion should be coupled inversely to the flux across the tropopause, exerting a similar forcing as the downward control principle (Haynes et al 1991)." First, I fail to see any connection at all to the downward control principle. I guess what the authors are referring to is mass conservation? Second, there is a seasonal cycle in the mass of the lowermost stratosphere, captured by the term dM/dt mentioned above, which implies a time lag between the downward fluxes at 380 K and at the tropopause. It seems that a direct link is being proposed between the downward flux at 380 K and the upward flux at the tropopause, with some time lag that is somewhat unclear (3 or 4 months). However it is not obvious to me why such a link would be expected. Perhaps the adiabatic flux from Olsen et al. (2004) or Schoeberl (2004) could be used instead, which constitutes the upward mass flux component. It peaks around October-November in the NH and March-April in the SH. This seasonality is in agreement with Skerlak et al. (2014), who find maximum TST flux in November for the NH and March for the SH.

2) The vertical movement of the WMO tropopause plays a crucial role in cross-tropopause flux, and it has strong seasonality, rising in spring and lowering in fall. Hence, the seasonality of the mean age of air probably changes substantially in tropopause-relative coordinates. These coordinates are used for the observational campaign data analysis (Fig. 6) but not for the ClaMS results (Fig. 5). The influence of tropopause altitude seasonality on the extratropical lower stratosphere age of air seasonality should be discussed.

Specific comments

- P7L5-6: however, the area is different for each region (larger for the tropics)

- P18L2-5: Could you explain why the seasonality of the fractions is not included? Would it not be more realistic if they were included? Otherwise why are they introduced for?

- P19L24-25: This sentence seems completely speculative. Please justify or remove.

- P21L10-15: Could it also be that isentropic transport around the subtropical jet is identified some times as tropical and other as extratropical, since the tropopause break is located at 30°N/S? In this case it would not be surprising that both tropical and extratropical spectra present recent flushing.

Technical corrections

- P2L10: succumbs - > presents, undergoes? (also on P22L12)

- P3L24: radioactive tracers is sometimes written with "" and sometimes not. Please uniformize.

- P6L6: remove comma

- P8L5: inhibits - > prevents

- P8L7: 'the division of both fluxes in these seasons' - > the ratio of fluxes in these two seasons

- P8L10: 'should be coupled inversely to the flux across the tropopause' - > to the upward flux across the tropopause (see general comment 1)

- P8L20: 'feedback' - > connection?

- P8L30: 'resemble' - > correspond to approximately?

- P10L7: 'transit time gradient of the mixing ratio' - > dependence?

- P15L29: 'maximum of downward forcing' - > maximum of downward transport

- P16L21: inhibit - > reduce / avoid

- P18L20: trends - > seasonal departures

- P18L24: with fresh tropospheric air

- P19L30: remove vice versa

- P20L23: what do you mean by 'finite datapoints'?

- P21L2-4: It would be useful to remind the reader the seasons in which each campaign took place

- P22L27: features - > provides

- P25L28-29: This sentence is unclear.

---

## Referee Comment (RC2) · Eric Ray (Referee) · 29 Apr 2020

This paper extends the work of a previous study by the same lead author on deriving age of air spectra in the stratosphere based on modeled and measured trace gases. The primary addition in this study is a refined treatment of the extratropical lowermost stratosphere by the inclusion of upward extratropical cross-tropopause entry into the stratosphere. This aspect of transport has been known for some time but this is the first study that has included non-tropical tropopause upward mass flux to obtain age spectra based on trace gas measurements in the NH lowermost stratosphere. This is important because most of the stratospheric in situ trace gas measurements we have

available now, and likely in the future, are in the lowermost stratosphere.

Overall this paper is a really nice piece of work. This study combined with the previous one by the same lead author has significantly advanced our ability to derive many aspects of the age of air and transport in the lower stratosphere from trace gas measurements. The use of the CLaMS model output to inform and validate the results derived from the trace gases is excellent and helps to understand the strengths and limitations of the inverse technique.

I recommend publication in ACP with consideration of the specific comments listed below. I have two main issues with the paper that should be easily resolved. One is the overall length of the text is too long and there are numerous grammatical errors such that I likely didn't find them all. The second issue is the discussion of the seasonal scaling that needs an additional figure and some more clarity in the text.

Specific comments:

Pg. 1, line 13: I have a preference to use 'output' when referring to models rather than 'data'. This clearly delineates the observational data from model output.

Pg. 1, line 29: add 'the' before 'extratropical', remove 'has' before 'peaked'

Pg. 1, line 30: 'The ratio of moments for all retrieved age spectra for PGS and WISE is found to range between 0.52 years and 2.81 years.'

Pg. 1, line 31: 'We conclude that. . .'

Pg. 2, line 2: '. . .stratosphere are determined by the global mean. . .'

Pg. 2, line 3: add 'the' before 'Brewer'

Pg. 2, line 6: 'recognized'

Pg. 2, line 13: change 'succumbs' to 'has' or 'shows'

Pg. 2, line 19: '. . .BDC will strengthen due to enhanced wave drag. . .'

Pg. 2, line 29: add 'the' before 'strength'

Pg. 2, line 33: remove 'linked'

Pg. 3, line 9: 'The basis of many past studies has been measurements...'

Pg. 3, line 19: change 'matching' to 'matched'

Pg. 3, line 22: 'We extend the inverse method described therein to the...'

Pg. 4, line 20: change 'an' to 'a'

Pg. 5, line 12: add 'a' after 'as'

Pg. 5, line 14: add comma after 'stratosphere'

Pg. 5, line 15: change 'steers' to 'affects'

Pg. 6, line 1: '...referring to transport through the tropopause section i.'

Pg. 7, line 5: change 'extent' to 'extend', add comma after 'choice'

Pg. 7, line 14: When you say 'now with 0.1% tolerance' does that mean in comparison to your previous study?

Pg. 7, line 28: In equation 13 why did you switch the symbol for the seasonal scaling from S in your previous paper to omega here? Just curious since I kind of liked S to stand for seasonal or scaling.

Section 2.2.2: I got unexpectedly hung up on this section even though I thought I knew what the scaling should look like. Your discussion of the seasonal scaling in the 2019 paper for tropical tropopause entry was very clear and I was expecting something similar here. After reading this a few times and staring at Figure 1 compared to your 2019 paper discussion I think I identified a couple of things that are missing here that would help. The main one is something equivalent to Figure 1a from your 2019 paper. You really need the visual of the mass flux seasonal cycle in each hemisphere to help make quick sense of the seasonal scalings. In this case the inverse Olsen flux would

likely be most appropriate. You might even want to include a latitude-height schematic of some kind. In the caption of Figure 1 you also need the sentence you had in the caption of Figure 1 from your 2019 paper, 'Increasing transit time means backward in time.'

You mention a couple of times here that the maximum scaling is in late spring and the minimum in late fall. And yet, we know that the maximum upward mass flux in the extratropics is in the summer and fall. Later in the paper it does seem to work out that you get peaks in the spectra in summer and fall but I don't follow how that works from this discussion. In your discussion of the tropical entry scaling it was clear that upward mass flux peaked in winter and there was a corresponding peak in the scaling curves. I'm left not confident that I understand this discussion and the scaling curves very well.

I would recommend rethinking this section from the point of view of a reader who hasn't read your 2019 paper and the seasonal scaling is all new.

Pg. 9, line 9: add 'a' after 'as'

Pg. 10, line 15: add 'a' after 'as'

Pg. 10, line 19: change 'perturbate' to 'perturb', add 'a' after 'As'

Pg. 10, line 31: change 'strongest' to 'most'

Pg. 11, line 3: I'm not sure what 'weakly regard the effective character' means. I would reword it somehow.

Pg. 11, line 5: change to 'approximations'

Pg. 11, line 7: '…tropopause consists of 10%...'

Pg. 11, line 12: change 'get' to 'were'

Pg. 12, line 8: add 'a' after 'as'

Pg. 13, line 13: add 'the' before 'troposphere'

Pg. 13, line 20: 'programmed'

Pg. 15, line 5: change 'their' to 'each' and 'sections' to 'section'

Pg. 15, line 7: add 'a' after 'as'

Pg. 15, line 8: '...setup is consistent overall, as the fractions at each location sum up...'

Pg. 15, line 13: add '450 K' after 'Below'

Pg. 15, line 14: remove 'start to'

Pg. 15, line 16: Here and everywhere else I would recommend abbreviating southern and northern hemispheres to SH and NH. This will help shorten the text a bit.

Pg. 15, line 17: remove 'also'

Pg. 15, line 18: remove 'quite'

Pg. 15, line 27: add 'tropospheric' after 'fresh'

Pg. 15, line 28: '...maximum downward forcing through the 380 K level is...'

Pg. 15, line 31: change 'that' to 'which'

Pg. 16, line 19: change 'referring to the' to 'for'

Pg. 16, line 20: change 'section' to 'entry'

Pg. 16, line 27: remove 'that also'

Pg. 17, line 2: Why not adjust the seasonal scaling to better match the pulse secondary peaks?

Pg. 17, line 14: add 'an' after 'on', remove 'fairly' and 'as well'

Pg. 17, line 16: remove 'largely'
Pg. 18, line 4: change 'using always' to 'with'

Pg. 18, line 6: remove 'in general'

Pg. 18, line 10: change 'find' to 'found'

Pg. 18, line 11: '...bias both above and below a threshold of 1.5 years...'

Pg. 18, line 18: Did you mean red shading here? Add 'the' after 'on'

Pg. 18, line 20: change 'find' to 'found'

Pg. 18, line 22: '...the sign appears different.'

Pg. 18, line 24: add 'tropospheric' after 'fresh'

Pg. 18, line 28: remove minus sign in front of 30S.

Pg. 18, line 31: add 'a' after 'as'

Section 4.3.1: You should restate the tracers used in these observational age spectra inversions. It would also be very helpful to restate the seasons of each mission at the beginning of this section since that's the most critical element in comparing them.

Pg. 19, line 11: add 'entry' after both 'NH' and 'tropical tropopause'

Pg. 19, line 13: change 'appears' to 'appear', add comma after 'general'

Pg. 19, line 16: add 'regions' after tropopause

Pg. 19, line 19: '...scattered bins of mean age older than 3 years...'

Pg. 20, line 1: It would be interesting to see the tropical – NH ages, maybe in a third row of plots in Figure 6.

Pg. 20, line 9: add 'the' before 'season'

Pg. 20, line 23: change 'datapoints' to 'locations' or something similar

[Figure]

Pg. 20, line 30: '...WISE rapidly decreases after...'

Pg. 21, line 10: add 'is' after 'This'

Pg. 21, line 33: add 'the' before 'right'

Pg. 22, line 12: change 'succumbs' to shows'

Pg. 22, line 13: add 'a' after 'as'

Pg. 22, line 30: change 'gained' to 'measured'

Section 5: The first nearly three pages of this section could be shortened considerably. The text of the paper is already quite long and the summary does not need to be so detailed. Just include the main points so it's easier for the reader to get the take home messages. In general, I would look for ways to shorten the text throughout the paper, it's a pretty long read.

Figures 6 and 7: Add somewhere prominently a label of the season of each mission since that's the most relevant comparison to be made between the plots.

---

## Author Comment (AC1) · 8 Jun 2020

We thank both referees for their comments on our manuscript.

Please find our detailed answers and a mark-up version of our manuscript in the attachment.

Please also note the supplement to this comment:
https://www.atmos-chem-phys-discuss.net/acp-2020-167/acp-2020-167-AC1-supplement.pdf

---

## Author Response (AR1)

We would like to thank both referees for their constructive and helpful comments on our manuscript, which have led to significant improvements. Specifically, we have revised the following parts of the manuscript:

- We have completely rethought the derivation of the extratropical seasonal scaling factors and revised most parts of Sect. 2.2.2. We have also carefully reread our cited previous studies on exchange processes between the troposphere and stratosphere in the extratropics to provide a now consistent and comprehensive evaluation.

- With the new scaling factors, which are now based on integrated CLaMS model output, all inversion procedures have been rerun and the respective figures (6, 7, and 8 as well as S1, S2, S3, and S4) and sections (4.2 and 4.3) have been adapted accordingly. The new factors are very similar to the previous ones so that no major changes of results and conclusions occurred.

- We have shortened the summary and removed redundant pieces of information to highlight the main aspects of our study. The conclusion/discussion part has not been shortened to stress the limitations of the method for the reader explicitly.

Changes are explained in detail below, where we answer each referee point by point. Referees' comments are shown in normal font, our answers in italic and changes to the manuscript in red.

**Answer to Eric Ray (Referee #2)**

This paper extends the work of a previous study by the same lead author on deriving age of air spectra in the stratosphere based on modeled and measured trace gases. The primary addition in this study is a refined treatment of the extratropical lowermost stratosphere by the inclusion of upward extratropical cross-tropopause entry into the stratosphere. This aspect of transport has been known for some time but this is the first study that has included non-tropical tropopause upward mass flux to obtain age spectra based on trace gas measurements in the NH lowermost stratosphere. This is important because most of the stratospheric in situ trace gas measurements we have available now, and likely in the future, are in the lowermost stratosphere.

Overall this paper is a really nice piece of work. This study combined with the previous one by the same lead author has significantly advanced our ability to derive many aspects of the age of air and transport in the lower stratosphere from trace gas measurements. The use of the CLaMS model output to inform and validate the results derived from the trace gases is excellent and helps to understand the strengths and limitations of the inverse technique.

I recommend publication in ACP with consideration of the specific comments listed below. I have two main issues with the paper that should be easily resolved. One is the overall length of the text is too long and there are numerous grammatical errors such that I likely didn't find them all. The second issue is the discussion of the seasonal scaling that needs an additional figure and some more clarity in the text.

*We thank Eric Ray for his positive assessment of our manuscript and appreciate his recommendation very much. We also want to thank him for his extensive specific comments that have led to major improvements of our manuscript. We have revised our section about the seasonal scaling for the defined tropopause regions completely by following the suggestion of Eric Ray to provide a now coherent reasoning with respect to previous studies. The inversion process has then been rerun for both the CLaMS output and observational data. The results sections and conclusions with respect to seasonality of the entrainment have been adapted accordingly. The newly derived and now thoroughly reasoned scaling factors are very similar to the ones in the previous version of the manuscript so that no major changes of the conclusions appear. The seasonal scaling is explained in more detail to ease the*

*reasoning for the reader, especially if they have not read our previous study (please see the specific comments below for details).*

*We have furthermore tried to shorten the manuscript without losing too much information. Especially the summary has been trimmed and redundant conclusions from the results section have been removed. However, since the section about the seasonal scaling (Sect. 2.2.2) is now longer than before, the overall length of the manuscript has only decreased slightly.*

Specific comments:

Pg. 1, line 13: I have a preference to use 'output' when referring to models rather than 'data'. This clearly delineates the observational data from model output.

*We have carefully replaced all "data" with "output" when referring to models in the manuscript, except for the title where we decided to stick to the old wording to keep it shorter.*

Pg. 1, line 29: add 'the' before 'extratropical', remove 'has' before 'peaked'

*Both done.*

Pg. 1, line 30: 'The ratio of moments for all retrieved age spectra for PGS and WISE is found to range between 0.52 years and 2.81 years.'

*Done.*

Pg. 1, line 31: 'We conclude that. . .'

*Done.*

Pg. 2, line 2: '. . .stratosphere are determined by the global mean. . .'

*Done.*

Pg. 2, line 3: add 'the' before 'Brewer'

*Done.*

Pg. 2, line 6: 'recognized'

*Done.*

Pg. 2, line 13: change 'succumbs' to 'has' or 'shows'

*We have changed it to "presents" following a suggestion by Referee #3.*

Pg. 2, line 19: '. . .BDC will strengthen due to enhanced wave drag. . .'

*Done.*

Pg. 2, line 29: add 'the' before 'strength'

*Done.*

Pg. 2, line 33: remove 'linked'

*Done.*

Pg. 3, line 9: 'The basis of many past studies has been measurements. . .'

*Done.*

Pg. 3, line 19: change 'matching' to 'matched'

*Done.*

Pg. 3, line 22: 'We extend the inverse method described therein to the. . .'

*Done.*

Pg. 4, line 20: change 'an' to 'a'

*Done.*

Pg. 5, line 12: add 'a' after 'as'

*Done.*

Pg. 5, line 14: add comma after 'stratosphere'

*Done.*

Pg. 5, line 15: change 'steers' to 'affects'

*Done.*

Pg. 6, line 1: '. . .referring to transport through the tropopause section i.'

*Done.*

Pg. 7, line 5: change 'extent' to 'extend', add comma after 'choice'

*Both done.*

Pg. 7, line 14: When you say 'now with 0.1% tolerance' does that mean in comparison to your previous study?

*That is correct. We have added the tolerance value of the previous study to stress that fact:*

"[…] now with 0.1 % tolerance (5 % in Hauck et al. (2019)) […]"

Pg. 7, line 28: In equation 13 why did you switch the symbol for the seasonal scaling from S in your previous paper to omega here? Just curious since I kind of liked S to stand for seasonal or scaling.

*The choice to change the variable symbol of the scaling factor from S to $\omega$ in this study was solely made to avoid inappropriate naming when the subscript of the southern hemispheric entry region is used together with the variable (i.e., $S_S$).*

Section 2.2.2: I got unexpectedly hung up on this section even though I thought I knew what the scaling should look like. Your discussion of the seasonal scaling in the 2019 paper for tropical tropopause entry was very clear and I was expecting something similar here. After reading this a few times and staring at Figure 1 compared to your 2019 paper discussion I think I identified a couple of things that are missing here that would help. The main one is something equivalent to Figure 1a from your 2019 paper. You really need the visual of the mass flux seasonal cycle in each hemisphere to help make quick sense of the seasonal scalings. In this case the inverse Olsen flux would likely be most appropriate. You might even want to include a latitude-height schematic of some kind. In the caption of Figure 1 you also need the sentence you had in the caption of Figure 1 from your 2019 paper, 'Increasing transit time means backward in time.'

You mention a couple of times here that the maximum scaling is in late spring and the minimum in late fall. And yet, we know that the maximum upward mass flux in the extratropics is in the summer and fall. Later in the paper it does seem to work out that you get peaks in the spectra in summer and fall but I don't follow how that works from this discussion. In your discussion of the tropical entry scaling it was clear that upward mass flux peaked in winter and there was a corresponding peak in the scaling curves. I'm left not confident that I understand this discussion and the scaling curves very well.

I would recommend rethinking this section from the point of view of a reader who hasn't read your 2019 paper and the seasonal scaling is all new.

*This is a very helpful comment that matches well with general comment 1) by Referee #3 (see below). Indeed, our derivation of the scaling factor and the reasoning with the results from previous work is not consistent. Additionally, as criticized by Referee #3, the choice of the inverted Olsen flux is quite arbitrary and not necessarily physically valid. We have therefore reconsidered the complete derivation process of the seasonal scaling factors and carefully reread our cited literature to provide a coherent argumentative structure.*

*While different observational studies (e.g. Bönisch et al. (2009)) indicate a flushing of the NH lowermost stratosphere during summer and fall, different studies of the hemispherically integrated troposphere to stratosphere mass flux across the tropopause indicate that the upward component of that flux reaches its maximum in (late) fall (Olsen et al., 2004; Schoeberl, 2004; Škerlak et al., 2014). Moreover, the net direction of that hemispherically integrated mass flux is downward. This strongly inhibits a scaling approach based on the related mass fluxes as the corresponding maxima and minima would appear at wrong transit times in the derived inverse age spectra.*

*Following this contradiction, we concluded that the hemispherically integrated mass flux is no suitable proxy for the upward transport across the defined extratropical tropopause segments. Instead, it is very likely that a narrow region around the subtropical jet stream at the subtropical borders of the NH/SH tropopause region controls the entrainment. This is in accordance with the findings of Yang et al. (2016) that identify a small region of net upward transport in the subtropics with a maximum in summer and a minimum in winter in both hemispheres. This also coincides robustly with the timing indicated by the observational study above. Therefore, for the setup in our study this appears to be the driving transport mechanisms that are also visible in the peaks of the CLaMS pulse age spectra. We have included this discussion into Sect. 2.2.2 as follows:*

"[…] The extratropical cycles are more challenging as distinct transport processes superimpose in the extratropical lowermost stratosphere. For a proper scaling factor in these regions, a net upward directed mass flux should be considered that reflects the ongoing dynamical processes as precisely as possible. Previous observationally based studies of $SF_6$, $CO_2$, and mean AoA find a flushing of the NH lowermost stratosphere with fresh tropospheric air during summer (JJA) and autumn (SON) that is most likely linked to the weaker subtropical jet stream and a dominance of the shallow branch of the BDC during that time (Bönisch et al., 2009). In contrast to these results, different mass budget analyses of the lowermost stratosphere in both hemispheres show that the net direction of the hemispherically integrated mass flux across the tropopause is downward with a maximum during spring in each hemisphere and a generally weaker seasonality in the SH. The upward component of this net mass flux is shown to reach its maximum during fall and its minimum conversely in spring in each hemisphere (Olsen et al., 2004; Schoeberl, 2004). The contradicting seasonality patterns imply that a hemispherically integrated mass flux might not be a suitable proxy for upward transport across the defined extratropical tropopause sections in this study, especially since the net direction of this flux is downward. It is more likely that a geographically narrow section of the NH and SH tropopause with year-round net upwelling causes the modes of the age spectra. Yang et al. (2016) investigate the ozone flux across the tropopause with a different framework where regions of net up- and downwelling are distinguishable. Their results indicate that in a small region in the subtropics of each hemisphere (around the equatorward flank of the subtropical jet stream), net upward transport across the tropopause with a maximum in summer is present, while at higher latitudes the net direction of the flux turns downward with a maximum in spring or winter depending on the latitudinal range (see their Fig. 12). In the SH, the seasonality is found to be generally weaker. This matches the observational results for the NH mentioned above. As the subtropical jet region is partly included in the defined tropopause sections for this study (30° − 90° N/S), it is likely that the enhanced entrainment across the subtropical jet stream during summer is a key feature of transport visible in derived age spectra. Unfortunately, Yang et al. (2016) provide only an ozone flux in their study (see their Fig. 7a and 7b) and no mass flux for the desired region so that a different proxy must be found. […]"

*Unfortunately, Yang et al. (2016) give only an ozone flux for the specific region and no mass flux that would be required for a proper scaling. Therefore, we have decided to follow Eric Ray's suggestion below and approximate the seasonality from CLaMS output directly. We have applied the ansatz by Ploeger and Birner (2016) and used integrated CLaMS age spectra to estimate the seasonal cycle in entrainment. We have integrated all stratospheric*
5 *CLaMS age spectra bin-wise to compute the fraction of air that entered across the NH/SH tropopause regions per transit time bin. The globally cumulated fractions now provide a robust average statistic for the seasonality in entrainment and are shown in the new top row of Fig. 1 following also the suggestion of Eric Ray. The fractions were then used to perform a relative scaling as in Hauck et al. (2019). The scaling is now also explained in more detail to ease the process for the reader. This new ansatz is included into Sect. 2.2.2:*

[revised manuscript text omitted]

*With the new scaling factors, all subsequent inversions of CLaMS model output and PGS/WISE data have been rerun. As stated above, the scaling factors are similar to the old ones, but now solidly reasoned. Therefore, no major changes of results and conclusions appeared. We have adapted the results (Sect. 4.2 and 4.3), figures (Fig. 3 to Fig. 8 and Fig. S1 to Fig. S4), summary, and conclusions properly (in particular the given values in the text – please see the mark-up version of the manuscript below).*

*Finally, we now mention the entrainment in the subtropical jet region as an important feature of transport, which is most likely visible in all our data as a dominant entrainment process. This has been included in Sect. 5:*

"[…] The maximum of entrainment across the NH tropopause section is found in general around JJA and SON. That coincides with the results of Bönisch et al. (2009), who find an enhancement of quasi-isentropic mixing across the weak NH subtropical jet stream during NH summer and fall. The maximum of intrusion in the SH midlatitudes can be detected accordingly with a shift of six months and reduced strength compared to the north around DJF and MAM. However, these seasonality patterns are contrary to the findings of multiple studies of seasonality using the hemispherically integrated upward mass fluxes across the tropopause that indicate a maximum in late fall (Olsen et al., 2004; Schoeberl, 2004; Škerlak et al., 2014). Our results might be an indication that the NH and SH origin fractions and age spectra in CLaMS are steered primarily by the intrusion processes across the jet stream around the subtropical border of the defined source regions. It is likely that if the boundary region is confined to higher latitudes, the seasonality of the related quantities will change as well. […]"

Pg. 9, line 9: add 'a' after 'as'

*Done.*

Pg. 10, line 15: add 'a' after 'as'

*Done.*

Pg. 10, line 19: change 'perturbate' to 'perturb', add 'a' after 'As'

*Both done.*

Pg. 10, line 31: change 'strongest' to 'most'

*Done.*

Pg. 11, line 3: I'm not sure what 'weakly regard the effective character' means. I would reword it somehow.

*With "weakly regard the effective character" we want to stress that local lifetimes do not quantify all relevant depletion processes for a specific age spectrum effectively. We have rephrased this paragraph to clarify it for the reader. It now reads:*

"[...] Long-lived trace gases show the most difficulties when assessing the first guess. On the one hand, global stratospheric lifetimes are likely an overestimation, as they are derived by dividing the global atmospheric burden by the global stratospheric loss rate. Local lifetimes, on the other hand, quantify the strength of localized stratospheric sink processes and thus do not consider that the desired lifetimes must express all relevant chemical depletion effectively for a given age spectrum. Additionally, these lifetimes are in many cases derived from model simulations. [...]"

Pg. 11, line 5: change to 'approximations'

*Done.*

Pg. 11, line 7: '. . .tropopause consists of 10%...'

*Done.*

Pg. 11, line 12: change 'get' to 'were'

*Done.*

Pg. 12, line 8: add 'a' after 'as'

*Done.*

Pg. 13, line 13: add 'the' before 'troposphere'

*Done.*

Pg. 13, line 20: 'programmed'

*Done.*

Pg. 15, line 5: change 'their' to 'each' and 'sections' to 'section'

*Both done.*

Pg. 15, line 7: add 'a' after 'as'

*Done.*

Pg. 15, line 8: '. . .setup is consistent overall, as the fractions at each location sum up. . .'

*Done.*

Pg. 15, line 13: add '450 K' after 'Below'

*Done.*

Pg. 15, line 14: remove 'start to'

*Done.*

Pg. 15, line 16: Here and everywhere else I would recommend abbreviating southern and northern hemispheres to SH and NH. This will help shorten the text a bit.

*We have abbreviated appropriate occurrences of northern and southern hemisphere/hemispheric to NH and SH throughout the manuscript. The abbreviations are introduced at the beginning of Sect. 2.2.2 as:*

"[…] Exchange processes across the northern hemispheric (NH) and southern hemispheric (SH) extratropical tropopause each display a different seasonality than the transport through the tropical tropopause layer. […]"

Pg. 15, line 17: remove 'also'

*Done.*

Pg. 15, line 18: remove 'quite'

*Done.*

Pg. 15, line 27: add 'tropospheric' after 'fresh'

*Done.*

Pg. 15, line 28: '. . .maximum downward forcing through the 380 K level is. . .'

5  *Done.*

Pg. 15, line 31: change 'that' to 'which'

*Done.*

Pg. 16, line 19: change 'referring to the' to 'for'

 *Done.*

Pg. 16, line 20: change 'section' to 'entry'

15  *Done.*

Pg. 16, line 27: remove 'that also'

*We have rephrased the sentence to:*

"[…] This indicates that the seasonal scaling works properly as it cancels out on annual average. […]"

Pg. 17, line 2: Why not adjust the seasonal scaling to better match the pulse secondary peaks?

*This is an interesting aspect. Theoretically, we could indeed adjust the scaling factors until they perfectly recreate all higher order maxima and minima of the CLaMS pulse spectra. However, it is unlikely that this adjustment can be achieved in a uniform way for every inverse spectrum so that a spatially varying scaling factor is probably*
25  *required. In turn, the retrieved inverse age spectra would then depend directly on the corresponding CLaMS age spectra, which is something we want to avoid.*

*As stated above, we have completely revised the seasonal scaling factor where we have considered Eric Ray's suggestion here. We could not find a suitable proxy for the upward mass flux at the NH/SH tropopause section in previous studies, which could explain the occurring maxima and minima in the CLaMS age spectra robustly. Thus,*
30  *we have decided to follow the bin-wise integrated CLaMS age spectra approach explained above, where we managed to keep the balance between an independent retrieval of the inverse age spectra and matching higher order maxima and minima (see above). Additionally, we have compared the retrieved seasonality with previous*

*studies to check if results are physically meaningful and consistent. Although the agreement is still not perfect, we think that this is tenable for the retrieval of inverse age spectra from observations.*

*Since with the new scaling factor we expect matching higher order modes, we have removed the explicit comparison in the results section to avoid circular reasoning and replaced it with a more general statement about the agreement between CLaMS and the inverse method. It reads for the NH spectra:*

"[…] Although the scaling factor is derived from the seasonal cycle in CLaMS and thus is expected to produce matching modes, the amplitude of the monomodal inverse spectra must be well-retrieved as otherwise the scaling would lead to deviating peaks and troughs. […]"

*And also for the age spectra in the SH:*

"[…] Higher order maxima and minima of the inverse spectra are in good agreement with the pulse spectra, which is expected as for the NH spectra above. […]"

Pg. 17, line 14: add 'an' after 'on', remove 'fairly' and 'as well'

*All done.*

Pg. 17, line 16: remove 'largely'

*Done.*

Pg. 18, line 4: change 'using always' to 'with'

*Done. Note that the sentence has been slightly adjusted following a suggestion by Referee #3.*

"[…] Since all origin fractions undergo a distinct seasonality, which is not necessarily identical with the seasonality of the age spectra, the composite spectrum of CLaMS pulse spectra and inverse method is calculated for this specific comparison only with annual mean origin fractions […]"

Pg. 18, line 6: remove 'in general'

*Done.*

Pg. 18, line 10: change 'find' to 'found'

*Done.*

Pg. 18, line 11: '. . .bias both above and below a threshold of 1.5 years. . .'

*Done.*

Pg. 18, line 18: Did you mean red shading here? Add 'the' after 'on'

*Yes, done.*

5   Pg. 18, line 20: change 'find' to 'found'

*Done.*

Pg. 18, line 22: '. . .the sign appears different.'

*Done.*

Pg. 18, line 24: add 'tropospheric' after 'fresh'

*Done.*

Pg. 18, line 28: remove minus sign in front of 30S.

15   *Done.*

Pg. 18, line 31: add 'a' after 'as'

*Done.*

20   Section 4.3.1: You should restate the tracers used in these observational age spectra inversions. It would also be very helpful to restate the seasons of each mission at the beginning of this section since that's the most critical element in comparing them.

*We have included the substances and the seasons as follows:*

"[…] The focus of the following sections is on the application of the inverse method to in-situ measurements of
25   11 chemically active trace gases ($SF_6$, $N_2O$, CFC-12, Halon 1211, Halon 1301, $CH_3Br$, $CH_2Br_2$, $CHBr_3$, $CHCl_2Br$, $CHClBr_2$, and $CH_2ClBr$) taken during the two aircraft campaigns PGS (phase 1 in winter 2015/2016, phase 2 in early spring 2016) and WISE (fall 2017). […]"

Pg. 19, line 11: add 'entry' after both 'NH' and 'tropical tropopause'

30   *Done.*

Pg. 19, line 13: change 'appears' to 'appear', add comma after 'general'

*Both done.*

Pg. 19, line 16: add 'regions' after tropopause'

*Done.*

Pg. 19, line 19: '. . .scattered bins of mean age older than 3 years. . .'

*The sentence has been rephrased with "mean AoA" to be consistent with the manuscript.*

"[…] Mean AoA referring to the NH tropopause (top row) is found to show the largest values of all data during PGS phase 2 with scattered bins of mean AoA older than 3 years around 90 K and 75° N. […]"

Pg. 20, line 1: It would be interesting to see the tropical – NH ages, maybe in a third row of plots in Figure 6.

*This is a very good suggestion. We have added a third row to Fig. 6 where the absolute difference between the northern and tropical mean AoA is shown and adjusted the corresponding descriptive text in the results section accordingly:*

"[…] Figure 6 depicts cross sections of mean AoA from the normalized inverse age spectra referring to the NH entry (top row) and tropical tropopause entry (mid row) during PGS phase 1 (first column), PGS phase 2 (second column) and WISE (third column). The absolute difference between NH and tropical mean AoA is shown in the bottom row of the figure. […]"

*and*

"[…] Mean AoA referring to the NH tropopause is generally smaller than the tropical counterpart, with an average difference ranging from -0.3 years for WISE to -0.36 years for PGS phase 2 and -0.46 years for phase 1. The difference is smaller at lower latitudes and increases with latitude and distance from the tropopause (see bottom row). […]"

Pg. 20, line 9: add 'the' before 'season'

*Done.*

Pg. 20, line 23: change 'datapoints' to 'locations' or something similar

*We have used "bins".*

Pg. 20, line 30: '. . .WISE rapidly decreases after. . .'

*Done.*

Pg. 21, line 10: add 'is' after 'This'

*Done.*

5  Pg. 21, line 33: add 'the' before 'right'

*Done.*

Pg. 22, line 12: change 'succumbs' to shows'

*We have changed it to "presents" according to Referee #3.*

Pg. 22, line 13: add 'a' after 'as'

*Done.*

Pg. 22, line 30: change 'gained' to 'measured'

15  *Done.*

Section 5: The first nearly three pages of this section could be shortened considerably. The text of the paper is already quite long and the summary does not need to be so detailed. Just include the main points so it's easier for the reader to get the take home messages. In general, I would look for ways to shorten the text throughout
20  the paper, it's a pretty long read.

*We have shortened the summary part of this section and removed conclusions that were redundant with the results section. The discussion part has not been shortened as we think that the capabilities of the method as well as its limitations must be stated here for the reader. We have also rephrased and shortened several sentences throughout the manuscript to shorten it slightly. However, as mentioned above, the section about the seasonal*
25  *scaling factor (Sect. 2.2.2) is now considerably longer to explain the scaling in a comprehensive way so that the overall length of the manuscript is almost unchanged.*

Figures 6 and 7: Add somewhere prominently a label of the season of each mission since that's the most relevant comparison to be made between the plots.

30  *We have added the season of each campaign to Fig. 6, Fig. 7, and Fig. 8 in the manuscript, as well as, Fig. S1, Fig. S2, Fig. S3 and Fig. S4 in the supplement. We have also included the year of the campaign to be more precise with the season.*

**Answer to Anonymous Referee #3**

This study presents an extension of the inverse methodology in Hauck et al. (2019) to derive stratospheric age of air from mixing ratios of a set of tracers, including entry of air masses through the tropical and extratropical tropopause of both hemispheres with corresponding seasonality scaling factors. The methodology is in general
5   valid and a novel result is an important role of upward transport from the extratropical tropopause, which could help explain inconsistencies in previous age of air results in the lowermost stratosphere. The paper is well written, especially the methodology sections. However, some clarification is needed on the proposed processes behind the results before I can recommend publication, in particular there is some confusion regarding the seasonality of the mass flux from the extratropical tropopause.

10  *We would like to thank Referee #3 for their assessment of our manuscript and appreciate their constructive comments very much. We would also like to thank the referee particularly for the clarification of the different mass flux components. We recognize that our evaluation of the seasonality in the mass flux across the tropopause in the initial version of the manuscript is inconsistent and confusing, which is why we have decided to start from scratch and completely revised the derivation of the extratropical scaling factors in Sect. 2.2.2. We have also*
15  *carefully reread the cited literature and considered additional work to come up with a now thorough and consistent reasoning. With the newly derived scaling factors, all inversion procedures of CLaMS model output and observational data have been rerun. The new scaling factors are very similar to the ones in the previous version of the manuscript so that results and especially conclusions remain valid (please see below for a detailed description of all changes to the manuscript).*

**General comments**

1) The discussion of the seasonality of upward flux from the extratropical tropopause and the obtention of scaling factors is quite confusing. In Section 2.2.2 (Extratropical seasonal cycles) it is stated that the scaling factor obtained based on previous works implies maximum upward flux from the tropopause into the stratosphere in
25  spring, and minimum in fall (lines 23-26, ll27-32 page 8). In contrast, the paper results suggest the opposite seasonality, with maximum in SON for the NH (e.g. lines 7-8 page 23). Nevertheless, the authors state that their results agree with previous works (e.g. lines11-12 page 23).

Importantly, I believe there is a wrong interpretation of the results in Fig. 6 of Appenzeller et al. (1996), which show the 'net mass flux across the tropopause due to mass variation of the lowermost stratosphere alone', that
30  is, the dM/dt term in their Eq. 1. This flux is considered here as 'the net flux across the tropopause', which is then argued to change sign with season (P8L4-5). However, the net flux across the extratropical tropopause is shown in Fig. 8 of A96, and corresponds with the term Fout in their Eq. 1. This flux is downward (negative) year-long, as argued also by subsequent works (including Olsen et al. 2004 cited here, see their Figure 2).

The seasonal cycle of the scaling factors is obtained taking reciprocal values of the 380 K downward flux from
35  Olsen et al. (2004). This is justified by saying that "this downward motion should be coupled inversely to the flux across the tropopause, exerting a similar forcing as the downward control principle (Haynes et al 1991)." First, I fail to see any connection at all to the downward control principle. I guess what the authors are referring to is mass conservation? Second, there is a seasonal cycle in the mass of the lowermost stratosphere, captured by the term dM/dt mentioned above, which implies a time lag between the downward fluxes at 380 K and at the
40  tropopause. It seems that a direct link is being proposed between the downward flux at 380 K and the upward flux at the tropopause, with some time lag that is somewhat unclear (3 or 4 months). However it is not obvious to me why such a link would be expected. Perhaps the adiabatic flux from Olsen et al. (2004) or Schoeberl (2004) could be used instead, which constitutes the upward mass flux component. It peaks around October-November

in the NH and March-April in the SH. This seasonality is in agreement with Skerlak et al. (2014), who find maximum TST flux in November for the NH and March for the SH.

*This is a very helpful comment and we would like to thank Referee #3 for the clarification of the considered mass fluxes, especially dM/dt in Appenzeller et al. (1996) which we have misinterpreted. This point agrees also very well*
5  *with one specific comment by Eric Ray (Referee #2 – see above). We recognize that our choice to use an inverted version of the net downward mass flux across 380 K in Olsen et al. (2004) to perform the scaling is quite arbitrary and not necessarily physically valid (we indeed referred to mass conservation rather than the downward control principle). Furthermore, our reasoning with the findings of previous studies is inconsistent and must be reassessed.*

*We have followed the suggestion of Referee #3 and used the adiabatic flux, i.e. the upward component of the*
10  *tropopause mass flux, given in Olsen et al. (2004) to perform a seasonal scaling for both hemispheres and repeated the CLaMS proof of concept with this new version. However, resulting inverse age spectra show higher order peaks and troughs that significantly deviate from the CLaMS pulse spectra especially during winter and spring. The peaks of the CLaMS pulse age spectra indicate that a maximum of entrainment across the defined extratropical tropopause section occurs in summer, while a minimum is visible in winter. This is contrary to the findings of Olsen*
15  *et al. (2004), Schoeberl (2004), and also Škerlak et al. (2014), who all show a maximum in hemispherically integrated TST in fall and a minimum in spring in both hemispheres.*

*With this contradiction we concluded that for the tropopause sections in our study, the hemispherically integrated TST is no suitable proxy for the scaling factors since the net direction is still downward. Instead, it is very likely that a small region of constant net upward motion is controlling the entrainment across the NH and SH extratropical*
20  *tropopause. Yang et al. (2016) identify a small region of net upward motion around the subtropical jet stream in each hemisphere (see their Fig. 12), which is at the tropical border of the defined NH and SH tropopause section. The maximum of upward transport in that region is consistently found during summer. For the NH, this matches with the results of Bönisch et al. (2009) that show a flushing of the northern lowermost stratosphere across the weak subtropical jet stream during summer and fall. Thus, the most suitable proxy would be the net mass flux*
25  *across the tropopause in the jet region, which is unfortunately not provided by Yang et al. (2016). This discussion is now included in Sect. 2.2.2:*

"[...] The extratropical cycles are more challenging as distinct transport processes superimpose in the extratropical lowermost stratosphere. For a proper scaling factor in these regions, a net upward directed mass flux should be considered that reflects the ongoing dynamical processes as precisely as possible. Previous observationally based
30  studies of $SF_6$, $CO_2$, and mean AoA find a flushing of the NH lowermost stratosphere with fresh tropospheric air during summer (JJA) and autumn (SON) that is most likely linked to the weaker subtropical jet stream and a dominance of the shallow branch of the BDC during that time (Bönisch et al., 2009). In contrast to these results, different mass budget analyses of the lowermost stratosphere in both hemispheres show that the net direction of the hemispherically integrated mass flux across the tropopause is downward with a maximum during spring in
35  each hemisphere and a generally weaker seasonality in the SH. The upward component of this net mass flux is shown to reach its maximum during fall and its minimum conversely in spring in each hemisphere (Olsen et al., 2004; Schoeberl, 2004). The contradicting seasonality patterns imply that a hemispherically integrated mass flux might not be a suitable proxy for upward transport across the defined extratropical tropopause sections in this study, especially since the net direction of this flux is downward. It is more likely that a geographically narrow
40  section of the NH and SH tropopause with year-round net upwelling causes the modes of the age spectra. Yang et al. (2016) investigate the ozone flux across the tropopause with a different framework where regions of net up- and downwelling are distinguishable. Their results indicate that in a small region in the subtropics of each hemisphere (around the equatorward flank of the subtropical jet stream), net upward transport across the tropopause with a maximum in summer is present, while at higher latitudes the net direction of the flux turns
45  downward with a maximum in spring or winter depending on the latitudinal range (see their Fig. 12). In the SH,

the seasonality is found to be generally weaker. This matches the observational results for the NH mentioned above. As the subtropical jet region is partly included in the defined tropopause sections for this study (30° – 90° N/S), it is likely that the enhanced entrainment across the subtropical jet stream during summer is a key feature of transport visible in derived age spectra. Unfortunately, Yang et al. (2016) provide only an ozone flux in their study (see their Fig. 7a and 7b) and no mass flux for the desired region so that a different proxy must be found. […]"

*Since Yang et al. (2016) focused on the ozone flux in the jet region, which is not necessarily equal to the corresponding mass flux, a suitable proxy had to be determined. This desired proxy had to quantify the net upward motion across the defined boundaries in our manuscript as precisely as possible. The best compromise was to use the CLaMS age spectra data from our TpSim simulation. We followed the suggestion of Eric Ray (see above) and used CLaMS model output to approximate the seasonality in entrainment. We have applied the ansatz by Ploeger and Birner (2016) and integrated monthly CLaMS age spectra bin-wise to estimate the percentage of air that enters across the boundary region of the age spectra per transit time bin. The transit time was then matched against real time. This strategy was done for all global stratospheric pulse age spectra in cumulative fashion to retrieve an average measure of the seasonality. The seasonal amount fractions were then used to perform the relative scaling as in Hauck et al. (2019), which is now also explained in more detail. The new amount fractions as well as the resulting scaling factors for the age spectrum are now shown in the new version of Fig. 1. The ansatz is explained in Sect. 2.2.2:*

"[…] To estimate the seasonality and the strength of the dominant entrainment processes specifically across the introduced extratropical tropopause sections, the modelled age spectra from the CLaMS simulation below are considered, which are initialized in the specified NH and SH tropopause section (see section 3.1 for details on the simulation). We follow the ansatz of Fig. 14b in Ploeger and Birner (2016) and integrate all monthly stratospheric age spectra of one source region bin-wise to compute the fraction of air that entered the stratosphere across this given source region per transit time bin. The fractions of all age spectra are cumulated and transit times are matched correctly against real time so that an average statistic for air mass entrainment across the NH and SH tropopause section per month is retrieved. Results of this ansatz are shown in the top row of Fig. 1 for the NH (panel (a)) and SH (panel (b)) tropopause section. It is evident that in the model for both regions the strongest entrainment occurs during July (NH) and January (SH) respectively, where more than 14 % of all air masses that cross the respective tropopause section are found to enter the stratosphere. This seasonality follows the observations of Bönisch et al. (2009) and also the ozone flux of Yang et al. (2016) very well and makes the subtropical jet region the most likely source mechanism for the tropopause sections defined above. The minimum of entrainment is found consistently in December (NH) and June (SH) with a fraction of less than 3 %. The cumulated values for each season are used to derive a scaling factor for the age spectra referring to the NH and SH tropopause sections. For instance, the fraction during JJA in the NH (ca. 39 %) is approximately three times larger than during DJF (ca. 13 %) so that corresponding age spectra in DJF must be tripled at transit times that correspond to JJA (0.5 years, 1.5 years, etc.). This principle is repeated for all remaining combinations of seasons in the NH and SH to estimate the coefficients in Eq. ( 13 ). No scaling is applied at transit times that represent the season the age spectrum is derived in, e.g., DJF in the example above. Resulting coefficients are shown in Tab. 1 and the final scaling factors are exemplified for the first year of transit time in the bottom row of Fig. 1. The scaling works consistently as the maximum of each curve is found at summer transit times while the minimum is located consistently during winter. […]"

*The use of CLaMS output for the derivation of the scaling factors is critically discussed in the manuscript. On the one hand, the usage limits the independence of the inverse method for the proof of concept since we now expect to gain maxima and minima at matching transit times. On the other hand, it is the most precise strategy to derive the seasonality for the specific setup in our study. Since the seasonality matches well with the ozon flux of the corresponding subtropical jet region in Yang et al. (2016) as well as with the observational studies above, which*

*indicate a maximum entrainment during summer and partly fall, the CLaMS results appear robust. Additionally, we have used all global stratospheric age spectra in integrated and cumulative fashion so that any information about the shape of the age spectrum in CLaMS is lost and not transferred to the inverse method. This is also stated in section 2.2.2:*

5 "[…] The scaling factors are approximated from integrated CLaMS age spectra, which aggravates a comparison of higher order peaks between the CLaMS reference and inverted age spectra as these modes are expected to appear at matching transit times. However, all global CLaMS age spectra are integrated and cumulated so that the resulting seasonality of the fractions is an average measure and no information about the exact shape is transferred from CLaMS to the inverse method. All inverse age spectra in one specific season are moreover scaled

10 with the same factor globally, which implies that the intrinsic amplitude of the monomodal inverse spectra must be well-retrieved as otherwise the scaling would nevertheless lead to deviating modes. Since the discovered seasonality in entrainment is also in good agreement with the upward ozone flux in the subtropical jet stream region (Yang et al., 2016) and with the seasonality derived from observations in the NH (Bönisch et al., 2009), the derived scaling factors are deemed a robust estimator for the presented extended inverse approach with the

15 specified NH and SH tropopause sections. […]"

*With the new scaling factors, all subsequent applications of the inverse method (CLaMS model output and observational data) have been repeated and the corresponding results sections (4.2 and 4.3) and figures (Fig. 3 to Fig. 8 and Fig S1. To Fig. S4) have been properly adapted. The new and old scaling factors are quite similar so that no major changes of results and conclusions arise. This could also be seen in the previous version of the manuscript*

20 *where we retrieved inverse age spectra that were in good agreement with the CLaMS pulse spectra despite the incorrectly derived old scaling factors. For the proof of concept (Sect. 4.2.1) an exact comparison of the transit times at the peaks in the inverse age spectra is not useful anymore so that we have decided to replace it with a more general statement about the agreement of higher order modes:*

"[…] Although the scaling factor is derived from the seasonal cycle in CLaMS and thus is expected to produce

25 matching modes, the amplitude of the monomodal inverse spectra must be well-retrieved as otherwise the scaling would lead to deviating peaks and troughs. […]"

*And for the SH age spectra:*

"[…] Higher order maxima and minima of the inverse spectra are in good agreement with the pulse spectra, which is expected as for the NH spectra above. […]"

30 *In Sect. 5 we have also added a short conclusive statement considering the disagreement between our detected flushing of the NH lowermost stratosphere prior to WISE and PGS phase 1 and the hemispherically integrated TST mass fluxes in Olsen et al. (2004), Schoeberl (2004), and Škerlak et al. (2014). This is stated as follows together with a possible starting point for future studies:*

"[…] The maximum of entrainment across the NH tropopause section is found in general around JJA and SON.

35 That coincides with the results of Bönisch et al. (2009), who find an enhancement of quasi-isentropic mixing across the weak NH subtropical jet stream during NH summer and fall. The maximum of intrusion in the SH midlatitudes can be detected accordingly with a shift of six months and reduced strength compared to the north around DJF and MAM. However, these seasonality patterns are contrary to the findings of multiple studies of seasonality using the hemispherically integrated upward mass fluxes across the tropopause that indicate a maximum in late

40 fall (Olsen et al., 2004; Schoeberl, 2004; Škerlak et al., 2014). Our results might be an indication that the NH and

SH origin fractions and age spectra in CLaMS are steered primarily by the intrusion processes across the jet stream around the subtropical border of the defined source regions. It is likely that if the boundary region is confined to higher latitudes, the seasonality of the related quantities will change as well. […]"

*Finally, we have included a short critical statement in Sect. 5 about the derivation of the seasonal scaling factors from CLaMS data to state that these might be only valid for the specific extratropical tropopause sections in our study. These could be reassessed in future studies as well with differently defined sections:*

"[…] Additionally, the scaling factors are derived from integrated CLaMS output and thus particularly created for the specific tropopause sections in this study. Although the seasonality matches results in previous work quite well and indicates that the subtropical jet stream is likely a dominant source region, it is likely that the retrieved scaling factors must be changed if the boundaries of the sections are shifted. Future studies could reassess these results using model output from other model simulations or differently defined NH and SH tropopause sections. […]"

2) The vertical movement of the WMO tropopause plays a crucial role in crosstropopause flux, and it has strong seasonality, rising in spring and lowering in fall. Hence, the seasonality of the mean age of air probably changes substantially in tropopause-relative coordinates. These coordinates are used for the observational campaign data analysis (Fig. 6) but not for the ClaMS results (Fig. 5). The influence of tropopause altitude seasonality on the extratropical lower stratosphere age of air seasonality should be discussed.

*The referee is correct. The vertical movement of the tropopause, especially the WMO tropopause, is a crucial factor for the seasonality of mean AoA and should thus be included for a precise evaluation of mean AoA. However, our main point in Fig. 5 of the manuscript is to qualitatively compare the performance of the inverse method with the CLaMS pulse spectra (and mean AoA) on the annual and seasonal scale. This is contrary to the focus of Fig. 6, where we explicitly analyze the spatial distribution and seasonal features of mean AoA and therefore switched to a tropopause relative coordinate system. Since CLaMS pulse and inverse mean AoA are both given in absolute coordinates, the variable tropopause height should affect both rows of the plot equally. This does not influence the validity of the proof of concept. Additionally, we have excluded the first 30 K in the stratosphere in all data to account for the tropospheric character of that region. This should dampen the effect of the varying WMO tropopause to some extent and also keeps our Fig. 6 easily comparable to Fig. 8 in Hauck et al. (2019), which is also given in absolute coordinates (yet pressure rather than potential temperature). We have included a short statement about that point into the manuscript in Sect. 4.2.2:*

"[…] Although a tropopause-relative coordinate system is generally preferable for an analysis of mean AoA close to the tropopause to incorporate the variable tropopause height throughout the year, absolute coordinates are chosen for this comparison to ease comparability with Fig. 8 in Hauck et al. (2019). Changes in tropopause height should affect the data in both rows of the figure similarly so that a comparison between CLaMS pulse and inverse mean AoA is not inhibited. […]"

Specific comments

- P7L5-6: however, the area is different for each region (larger for the tropics)

*That is correct and a very important aspect. We now mention that fact explicitly:*

"[…] With that choice, all entry regions span an identical range of 60° latitude, although the actual enclosed area is larger for the tropical section. […]"

- P18L2-5: Could you explain why the seasonality of the fractions is not included? Would it not be more realistic if they were included? Otherwise why are they introduced for?

*The seasonality of the fraction is only excluded for this specific comparison in the proof of concept, as we are primarily interested in the seasonality of mean AoA and not in that of the origin fraction. As a decrease of mean AoA is often accompanied by an increase of the respective origin fraction, the seasonality of mean AoA could be masked by that of the origin fraction.*

*However, at the same time we want to evaluate the general performance of all three age spectra (tropical, NH, SH) combined at any stratospheric location weighted by their relative importance at that location, which is provided by the origin fraction. We tried to avoid the interference of the seasonality of mean AoA and origin fractions and preserve the weighting by using the annual mean origin fractions instead. This allows us to create a figure that can be easily compared to Fig. 8 in our previous study and detect improvements at a glance.*

*We state that now more clearly in Sect. 4.2.2:*

"[…] A seasonal analysis of the composite spectrum is advantageous to assess the behavior of all three different age spectra – northern, tropical, and southern – simultaneously, but weighted by their geographical importance. Since all origin fractions undergo a distinct seasonality, which is not necessarily identical with the seasonality of the age spectra, the composite spectrum of CLaMS pulse spectra and inverse method is calculated for this specific comparison only with annual mean origin fractions in Eq. ( 8 ) (inserted into Eq. ( 6 )). This ensures that the presented seasonal differences are only steered by the inverted age spectra and preserves the weighting of the individual age spectra at the same time. […]"

- P19L24-25: This sentence seems completely speculative. Please justify or remove.

*We have removed the sentence.*

- P21L10-15: Could it also be that isentropic transport around the subtropical jet is identified some times as tropical and other as extratropical, since the tropopause break is located at 30◦N/S? In this case it would not be surprising that both tropical and extratropical spectra present recent flushing.

*This is a very good suggestion. Indeed, the subtropical jet stream appears to play an important role in air mass entrainment across the northern (and also southern) extratropical tropopause section as defined in our manuscript. As stated above, we also find that the seasonality of entrainment in CLaMS across the northern/southern tropopause sections seems to follow the stronger and weaker phases of the jet in each hemisphere causing the maxima and minima of the corresponding pulse age spectra. This makes the jet stream a very important feature of entrainment in our study.*

*We have rephrased the paragraph and included the jet stream explicitly as a plausible cause of the entrainment across the tropical and northern tropopause section. It now reads:*

"[…] Possible causes might be the shallow branch of the BDC in proximity to the tropopause or the subtropical jet stream drifting around the border of the specified tropical and NH extratropical tropopause section that both could interfere with the seasonality of transport across the tropical tropopause. […]"

*And we also stress it again in the summary:*

"[…] Campaign-averaged inverse spectra indicate a strong unexpected intrusion across the tropical tropopause prior to WISE and PGS phase 1 that might be related to entrainment around the subtropical jet stream. […]"

Technical corrections

- P2L10: succumbs - > presents, undergoes? (also on P22L12)

*We have changed both to "presents".*

- P3L24: radioactive tracers is sometimes written with "" and sometimes not. Please uniformize.

*The formulation is now uniformized and the quotation marks have been removed.*

- P6L6: remove comma

*Done.*

- P8L5: inhibits - > prevents

*Since we have revised the concept of the extratropical scaling factor completely, the corresponding part of Sect. 2.2.2 has been rewritten. Therefore, the sentence is no longer present in the manuscript.*

- P8L7: 'the division of both fluxes in these seasons' - > the ratio of fluxes in these two seasons

*This sentence is also no longer included.*

- P8L10: 'should be coupled inversely to the flux across the tropopause' - > to the upward flux across the tropopause (see general comment 1)

*This sentence is also no longer included.*

- P8L20: 'feedback' - > connection?

*This sentence is also no longer included.*

- P8L30: 'resemble' - > correspond to approximately?

*This sentence is also no longer included.*

- P10L7: 'transit time gradient of the mixing ratio' - > dependence?

We have replaced the word "gradient" with "dependence".

- P15L29: 'maximum of downward forcing' - > maximum of downward transport

*The sentence has been rephrased following the suggestion of Eric Ray:*

"[…] Since the maximum downward forcing through the 380 K level is simulated in late January, the NH origin fraction attains its minimum in MAM. […]"

- P16L21: inhibit - > reduce / avoid

*We have replaced "inhibit" with "avoid".*

- P18L20: trends - > seasonal departures

*We have replaced "trends" with "seasonal differences".*

- P18L24: with fresh tropospheric air

*Done.*

- P19L30: remove vice versa

*Done.*

- P20L23: what do you mean by 'finite datapoints'?

*With finite datapoints we mean datapoints that are present in PGS phase 1, PGS phase 2, and also WISE. As this wording is not clear, we have rephrased it to:*

"[…] To ensure comparability, the campaign average is constructed by selecting only bins that are present in both PGS phases and WISE. […]"

- P21L2-4: It would be useful to remind the reader the seasons in which each campaign took place

*We have included the respective season of the campaign into the first sentence of Sect. 4.3.2:*

"[…] Figure 7 presents the campaign-averaged age spectra derived by the inverse method with reference at the NH tropopause (panel (a)) and tropical tropopause (panel (b)) for PGS phase 1 (DJF; blue), PGS phase 2 (MAM; green) and WISE (SON; orange). […]"

- P22L27: features - > provides

*This sentence has been removed from the manuscript to shorten the summary following Eric Ray's suggestion.*

- P25L28-29: This sentence is unclear.

*With this sentence we want to stress that the scaling factor repeats for every year of transit time so that seasons of unusually strong or weak entrainment are not visible in the maxima and minima of the retrieved inverse age spectrum. This is now clarified:*

[revised manuscript text omitted]